# Synergistic Crosstalk of PACAP and Notch Signaling Pathways in Bone Development

**DOI:** 10.3390/ijms26115088

**Published:** 2025-05-26

**Authors:** Vince Szegeczki, Andrea Pálfi, Csaba Fillér, Barbara Hinnah, Anna Tóth, Lili Sarolta Kovács, Adél Jüngling, Róza Zákány, Dóra Reglődi, Tamás Juhász

**Affiliations:** 1Department of Anatomy, Histology and Embryology, Faculty of Medicine, University of Debrecen, Nagyerdei krt. 98, H-4032 Debrecen, Hungary; szeg.vince@gmail.com (V.S.); p_andi@hotmail.hu (A.P.); filler.csaba@med.unideb.hu (C.F.); hinnah.barbara@med.unideb.hu (B.H.); tothanncz@gmail.com (A.T.); kovacs.klili@gmail.com (L.S.K.); roza@anat.med.unideb.hu (R.Z.); 2Department of Anatomy, HUN-REN-PTE PACAP Research Team, Medical School, University of Pécs, Szigeti út 12, H-7624 Pécs, Hungary; junglingadel@gmail.com (A.J.); dora.reglodi@aok.pte.hu (D.R.)

**Keywords:** DLL, Notch, ossification, inorganic matrix, CSL, NFATc1

## Abstract

Pituitary adenylate cyclase-activating polypeptide (PACAP) is a neuropeptide that plays significant regulatory roles in the differentiation of the central nervous system and peripheral organs. A lack of the neuropeptide can lead to abnormalities in long bone development. In callus formation, a possible signaling balance shift in PACAP KO mice has been demonstrated, but Notch signalization, with its potential connection with PACAP 1-38, has not been investigated in ossification. Our main goal was to show connections between PACAP and Notch signaling in osteogenesis. Notch signalization showed an elevation in the long bones of PACAP-gene-deficient mice, and it was also elevated during the PACAP 1-38 treatment of UMR-106 and MC3T3-E1 osteogenic cells. Moreover, the inhibition of Notch signaling was compensated by the addition of PACAP 1-38 in vitro. The inorganic and organic matrix production of UMR-106 cells was increased during PACAP 1-38 treatment under the inhibition of Notch signaling. As a possible common target, the expression and nuclear translocation of NFATc1 transcription factor was increased during the disturbance of PACAP and Notch signaling. Our results indicate a possible synergistic regulation during bone formation by PACAP and Notch signalization. The crosstalk between Notch and PACAP signaling pathways highlights the complexity of bone development and homeostasis.

## 1. Introduction

Bone is one of the hardest tissues in the body; it is essential for weight support and provides a skeletal framework for movement. Histologically, long bones have two parts: compact and spongy bone. Compact bone constitutes the outer layer of long bones, protecting the bone marrow, while its thickness contributes to the stability required for weight-bearing [1]. In the cortical bone, a specific arrangement of extracellular matrix molecules can be found: collagen type I forms concentric lamellae in the osteons, featuring a spiral structure that runs perpendicular to adjacent collagen fibers. This alignment is crucial for determining the shape and stability of long bones [2]. Calcium phosphate crystals are embedded in the gaps between collagen fibers, forming the inorganic component that provides mechanical strength to cortical bone [3]. Any pathological disruption in proper cortical bone formation during endochondral ossification can lead to an abnormal architecture that diminishes the mechanical stability of long bones [4]. Traumatic fractures in long bones and subsequent disintegration trigger the appropriate expression of the matrix along with organized cytokine secretion, resulting in a well-structured, stable matrix architecture and orientation. The healing process of cortical bone begins with local inflammation, followed by callus formation and the remodeling of both the organic and inorganic matrices [5]. Intramembranous ossification is the other essential process through which bone tissue is formed. It occurs within ossification centers: specifically, the mesenchyme membrane, which contains undifferentiated stem cells that differentiate into osteoprogenitor cells and then osteoblasts. The latter ones secrete an organic matrix, mainly comprising collagen fibers, which serve as the framework for bone formation. Calcium salts are then deposited onto the collagen fibers, gradually hardening the bone tissue. Intramembranous ossification primarily occurs in flat bones of the skull, facial bones and clavicles [6].

Bone formation and regeneration are intricately organized processes governed by various signaling pathways. Initially, osteoprogenitor cells proliferate rapidly before differentiating into early osteoblasts that secrete the organic bone matrix [7]. Following this, the activity of late osteoblasts leads to significant mineralization of the extracellular matrix (ECM), eventually resulting in the formation of osteocytes [7]. The differentiation of osteogenic cells is triggered by several key regulatory pathways, including the activation of bone morphogenetic proteins (BMP) [8], the wingless-related integration site (WNT) [9], fibroblast growth factor (FGF) [10] and hedgehog (HH) signaling [11], all of which are crucial for proper bone formation. The activation of BMP receptors through Smads can lead to increased expression of alkaline phosphatase (ALP) or collagen type I, as well as stimulating the expression of bone-specific transcription factors such as osterix [12]. One potential mechanism for the transcriptional activation of BMP-encoding genes is the enhanced activity of protein kinase A (PKA), which can phosphorylate the cAMP response element-binding protein (CREB) transcription factor, subsequently translocating it to the nucleus to induce the mRNA expression of BMPs [13].

Pituitary adenylate cyclase-activating polypeptide (PACAP) is a neuropeptide that plays a crucial role in various physiological processes [14], including the regulation of bone formation [15]. PACAP is known to exert its effects through binding to specific receptors, primarily PACAP type 1 receptors (PAC1), which are expressed in bone cells such as osteoblasts and osteocytes [16]. Our previous results indicate that PACAP promotes osteoblast differentiation and maturation, enhancing the production of key bone matrix proteins, such as collagen and osteocalcin [16]. PACAP’s influence on bone formation extends to its involvement in the response to mechanical loading [17]. Studies suggest that PACAP signaling can be activated by mechanical stress, leading to the increased expression of osteogenic markers [17,18]. This mechanism ensures that bone tissue can adapt and strengthen in response to physical demands, contributing to overall skeletal health. Moreover, PACAP’s anti-inflammatory properties may also play a role in its effects on bone. By mitigating inflammation, it can prevent the detrimental effects of inflammatory cytokines on bone cells, further supporting osteogenesis and protecting against conditions such as osteoporosis [19]. Furthermore, PACAP interacts with other signaling pathways, including those involving BMPs and WNT, enhancing its overall effects on bone formation [20]. It has already been shown that PACAP has a positive role in bone formation and differentiation [16] and regulates a well-balanced signaling in callus formation [21]. Moreover, the absence of the neuropeptide resulted in increased fragility and abnormal inorganic matrix production [15]. This crosstalk amplifies the osteogenic responses initiated by PACAP, creating a synergistic environment conducive to bone regeneration.

Notch signaling is a critical pathway that regulates various cellular processes, including differentiation, proliferation and apoptosis [22]. In the context of bone formation, Notch signaling exerts significant influence during several key stages of osteogenesis, affecting both the osteoprogenitor cell population and mature osteoblasts [23]. During embryonic development, the Notch pathway plays a fundamental role in regulating the balance between osteoblasts and adipocytes [24]. It helps to maintain the pool of osteoprogenitor cells by preventing their differentiation into adipocytes, thereby ensuring that enough cells are committed to bone formation. In addition to maintaining osteoprogenitor cells, Notch signaling also impacts the differentiation process of these cells into osteoblasts [24]. Activation of Notch receptors, particularly Notch1, can promote osteogenic differentiation by inhibiting the effect of negative regulatory signals [25]. For example, the Notch pathway is known to modulate the expression of bone morphogenetic proteins (BMPs) [26], which are essential for osteogenesis. BMPs, in turn, can reinforce Notch signaling, creating a positive feedback loop that enhances bone formation. Moreover, Notch signaling influences the maturation of osteoblasts. The pathway encourages the production of essential bone matrix proteins, such as collagen and osteocalcin [23]. Furthermore, it aids in the mineralization process, which is crucial for the structural integrity of long bones. The precise regulation of these processes by Notch ensures that osteoblasts proceed through their maturation stages effectively [27]. Notch signaling also plays a role in the regulation of osteocyte function, as it is intricately involved in bone remodeling and homeostasis, and it is believed to influence their survival and functional capacities [28]. This is vital, since osteocytes coordinate the activity of osteoblasts and osteoclasts, the latter being responsible for bone resorption. The interaction between Notch signaling and other pathways is another important aspect of its function in long bone formation [28]. For instance, there is evidence of crosstalk between the Notch and WNT signaling pathways, which also play a role in osteogenesis. This interaction can modulate the expression of key osteogenic genes, thereby reinforcing bone formation and remodeling [29]. Notch signaling plays a significant role in regulating intramembranous ossification as it influences the differentiation of mesenchymal cells into osteoblasts. Notch receptors on the surface of mesenchymal cells interact with Notch ligands, triggering a cascade of events that promotes the expression of genes that are essential for osteoblast development and function [30]. The dysregulation of Notch signaling has been associated with several bone-related disorders, including osteoporosis and osteosarcoma [25]. In osteoporosis, impaired Notch signaling may lead to excessive adipogenesis from osteoprogenitor cells and reduced osteoblast function [31], while, in osteosarcoma, aberrant Notch activity can promote tumorigenesis [32]. On the other hand, the lack of PACAP leads to the upregulation of Notch pathway elements in the odontoblast and ameloblast cells, also suggesting the pivotal role of PACAP in the regulation of hard tissue matrix formation [33]. Recent studies suggest that PACAP may interact with the Notch signaling pathway to modulate skeletal element formation. For instance, PACAP can enhance the expression of Notch receptors and ligands, potentially promoting tooth development [33]. Although we do not have data about their direct signaling crosstalk in bone formation, this interaction may help coordinate the balance between bone formation and resorption, contributing to overall bone health.

In summary, PACAP and Notch signaling are key regulators of bone formation, influencing osteoblast activity, but their direct signaling crosstalk has not been clarified yet. Investigating the precise mechanisms by which PACAP exerts its actions can provide valuable insights into potential therapeutic strategies for treating bone-related disorders and promoting skeletal health.

## 2. Results

### 2.1. Notch Signalling Alterations in Femurs of PACAP-Gene-Deficient Mice

The expression of Notch receptors was followed with RT-PCR reactions, and *Notch1*, *Notch2* and *Notch3* were detectable in WT mice, while the *Notch4* receptor mRNA was not identifiable (Figure 1A). On the other hand, in PACAP KO mice, the mRNAs of the receptors were barely detectable and the expression of the *Notch4* receptor was not shown (Figure 1A). The protein expression of these receptors showed a similar expression profile as that presented in WT femurs and their protein expression was diminished in PACAP KO mice (Figure 1B,C).

In mammals, there are several DLL proteins, including DLL1, DLL3, and DLL4, each with distinct functions and expression patterns in various tissues and developmental stages [22,34]. In femurs of WT mice, we were able to detect the mRNAs of all DLLs with various expression strengths. The *DLL1* showed a strong expression in WT animals and a further increase in mRNA expression was demonstrated in PACAP KO littermates (Figure 1A). *DLL3* and *DLL4* were at the level detection threshold in WT mice, but our results showed a significant elevation in PACAP KO femurs (Figure 1A). Protein expression of DLL1 and DLL3 was demonstrated in WT animals and, similarly to mRNA expression, elevated protein expression was shown in PACAP KO littermates (Figure 1B,C). The expression of DLL4 was barely detectable in the femurs of WT animals and a slight but significant increase was demonstrated in PACAP KO mice (Figure 1B,C).

The mRNA expression of *Jagged* ligands was detectable in WT femurs and showed a significant decrease in PACAP KO mice (Figure 1A). On the other hand, the protein expression of Jagged1 did not alter in gene-deficient mice (Figure 1B,C). Similarly to mRNA expression, the protein of Jagged2 ligand was diminished in PACAP KO littermates (Figure 1B,C).

The mRNA expression of *TACE* was detected in WT mice and a significant elevation was shown in PACAP KO littermates (Figure 1A). Furthermore, a significant increase was observed in the protein expression of TACE in PACAP KO femurs (Figure 1B,C).

By cleaving and releasing these ligands from the cell surface, ADAM9 can modulate the strength and duration of Notch signaling. In WT femurs, the mRNA and protein expression of ADAM9 was hardly detectable (Figure 1A–C). On the contrary, ADAM9 showed a significant elevation both in mRNA and protein levels in PACAP KO mice (Figure 1A–C).

The mRNA expression of *CSL* was shown to be equal in WT and PACAP KO mice (Figure 1A). On the contrary, the protein expression of CSL in PACAP KO femurs showed a strong elevation compared to WT mice (Figure 1B,C).

The mRNA expression of *Numb* was detectable in WT femurs, with a significant elevation in PACAP KO mice (Figure 1A). Protein expression of the inhibitor was also present in WT femurs and showed a strong increase in PACAP KO samples (Figure 1B,C).

### 2.2. Inhibition of Notch Signalling Is Compensated by PACAP 1-38 in Osteoblastic Cell Line

As the absence of PACAP altered the expression of Notch signaling elements, we aimed to study the effects of PACAP 1-38 on in vitro osteogenesis using the UMR-106 osteoblastic cell line, an appropriate model for osteoblastic differentiation in long bones. The activation of PACAP signaling and the inhibition of Notch signaling can clarify the crosstalk mechanism between the two pathways. On the fourth day of culturing, we aimed to observe the potential cytotoxic effects of PACAP 1-38 and DAPT treatments using a viability test. To evaluate the non-toxic concentration of DAPT in UMR-106 cells, 1 µM, 5 µM and 10 µM of DAPT were added to cell cultures after differentiation for four days. We found that 5 µM of DAPT did not decrease the viability of cells (Figure 2A). On the other hand, the administration of PACAP 1-38 and DAPT did not increase the mitochondrial activity of UMR-106 cells, but their combined application significantly elevated osteoblastic mitochondrial activity (Figure 2B).

Furthermore, we also assessed proliferation activity by examining the incorporation of tritiated thymidine, as both PACAP and Notch influence cellular division. As a result of adding PACAP 1-38, the incorporation rate of tritiated thymidine significantly increased (Figure 2C), as demonstrated in our previous results [16]. On the other hand, a slight but not significant elevation was detected in the presence of DAPT (Figure 2C). The combined application of PACAP 1-38 and DAPT elevated the proliferation, but it was not additive (Figure 2C).

The mRNA expression of *PKA*, *ALP*, and *Coll. Ia1* was also followed by conventional RT-PCR reactions. The administration of PACAP 1-38 did not alter *Coll. Ia1* expression (Figure 2D). Interestingly, adding DAPT increased the mRNA expression of *Coll. Ia1*, and the combined application of PACAP 1-38 and Notch inhibition resulted in an elevation (Figure 2D). As a direct target of PACAP signaling is PKA, we demonstrated that the mRNA expression was significantly increased in UMR cells in the presence of PACAP 1-38 (Figure 2D). On the other hand, the inhibition of Notch signaling did not alter the *PKA* mRNA expression, but the two effects were stronger (Figure 2D). Interestingly, neither PACAP 1-38 nor DAPT had any effect on the mRNA expression of *ALP* (Figure 2D).

The protein expression of collagen type I was strongly elevated in the presence of the neuropeptide and increased during the inhibition of Notch signaling (Figure 2E,F). Moreover, the combined treatment had an additive effect on collagen expression (Figure 2E,F). The protein expression of PKA showed a slight elevation in the presence of PACAP 1-38, but a significant reduction was detected during the inhibition of Notch signaling (Figure 2E,F). Interestingly, the combined treatment resulted in the doubled protein expression of PKA (Figure 2E,F). The protein expression of ALP was augmented by PACAP 1-38 and, to a lesser extent, by DAPT, but it was not additive in a combined treatment (Figure 2E,F). Therefore, we monitored the activity of ALP after the addition of PACAP 1-38 either individually or in combination with DAPT, which resulted in an increase in ALP activity (Figure 2G). A slight but not significant elevation was also detected in the inhibition of Notch signaling (Figure 2G).

The calcification processes in the presence of PACAP 1-38 and DAPT were followed by histological staining. Alizarin red staining was used for the detection of intra- and extracellular Ca^2+^. The addition of PACAP 1-38 resulted in mostly extracellularly accumulated Ca^2+^ (Figure 3A) and DAPT inhibitors that were mostly elevated the intracellular Ca^2+^ level of UMR-106 cells (Figure 3A). Subsequently, the combination of the treatments elevated the intra- and extracellular Ca^2+^ in parallel (Figure 3A). Moreover, Ca-phosphate crystal accumulation was also followed by von Kossa staining, which further strengthened the previous results, as PACAP elevated Ca-phosphate around the cells (Figure 3B). On the other hand, the inhibition of Notch signaling with DAPT did not alter the extracellular calcification process (Figure 3B). The combined treatment resulted in stronger Ca-phosphate crystal accumulation around the UMR-106 cells (Figure 3B).

### 2.3. Notch Signalling Showed Altered Expression After the Addition of PACAP 1-38 in Osteoblast Cell Lines

The expression of Notch receptors was followed with RT-PCR reactions, and *Notch1*, *Notch2* and *Notch3* were detectable in the UMR-106 cell line, similarly to findings in femurs (Figure 4A). On the other hand, no significant alteration was detected in the mRNA expression of *Notch1* and *Notch2* receptors after PACAP 1-38 administration or the addition of DAPT (Figure 4A). Interestingly, a slight reduction in *Notch3* mRNA expression was detected in the presence of both PACAP 1-38 and DAPT (Figure 4A). On the contrary, significant elevation was shown in the protein expression of Notch1 receptors in the presence of PACAP 1-38, which was decreased by the inhibition of Notch signaling (Figure 4B,C). Protein expression of the Notch2 receptor was not altered in the presence of the neuropeptide but elevated after DAPT administration, even when it occurred together with PACAP 1-38 administration (Figure 4B,C). Notch3 protein expression was elevated in the presence of DAPT but was reduced after the application of PACAP (Figure 4B,C). On the other hand, PACAP 1-38 was able to compensate for the effects of Notch inhibition and resulted in the normalization of Notch3 receptor protein expression (Figure 4B,C).

*DLL1* mRNA expression was significantly increased after DAPT administration, even in combination with PACAP 1-38, while the addition of PACAP 1-38 did not significantly increase its expression (Figure 4A). On the contrary, the protein expression of DLL1 significantly increased in the presence of the neuropeptide, but the inhibition of Notch signaling reduced the protein level of DLL1. Moreover, PACAP 1-38 was not able to compensate for the reducing effect of DAPT in the combined application (Figure 4B,C). *DLL3* mRNA expression did not significantly alter either with the presence of PACAP 1-38 or during the administration of DAPT, but the combined application resulted in a slight but not significant elevation (Figure 4A). Interestingly, the protein expression detection of DLL3 was on the detection limit of this method, but DAPT reduced its protein expression and PACAP 1-38 was partially able to compensate for this decrease in combined administration (Figure 4B,C). On the other hand, the mRNA expression of *DLL4* increased in the presence of PACAP 1-38 but showed a reduction after DAPT treatment. The combination of PACAP 1-38 and DAPT addition elevated the mRNA expression of *DLL4* (Figure 4A). Similarly, the protein expression of DLL4 was elevated by PACAP 1-38 and decreased by adding DAPT, while PACAP 1-38 compensated for the decreasing effect of DAPT treatment (Figure 4B,C).

Other ligands of Notch signaling, such as *Jagged1*, showed alterations, as the mRNA expression of this ligand significantly elevated both in the presence of PACAP 1-38 and DAPT (Figure 4A). On the contrary, the protein expression was elevated in the presence of PACAP 1-38 but was significantly reduced after the addition of DAPT, which was not compensated by the addition of PACAP 1-38 (Figure 4B,C). *Jagged2* mRNA expression was augmented by PACAP 1-38, but no alteration was detected upon adding DAPT, and PACAP 1-38 compensated the effects of Notch inhibition (Figure 4A). On the other hand, strong elevation was not detected in protein expression in the presence of PACAP 1-38, but the neuropeptide was able to compensate the protein-expression-reducing effect of DAPT (Figure 4B,C).

*TACE* mRNA expression did not alter in the presence of PACAP 1-38 or DAPT (Figure 4A), but strong protein expression was demonstrated during PACAP 1-38 administration (Figure 4B,C). Moreover, the neuropeptide was able to compensate for TACE protein reduction after DAPT addition (Figure 4B,C). *Adam9* mRNA expression was slightly elevated during the activation of PACAP signaling but reduced after the inhibition of Notch signaling. On the other hand, the combined treatment resulted in a significant elevation in the mRNA expression of *Adam9* (Figure 4A). Similarly, the protein expression was elevated in the presence of PACAP 1-38 but reduced after DAPT addition, and a further reduction was documented in the combined administration (Figure 4B,C). In *CSL* mRNA expression, similarly to TACE, no alteration was demonstrated after PACAP 1-38 or DAPT treatment (Figure 4A). On the other hand, the protein expression of CSL showed an elevation during PACAP 1-38 administration but was reduced by DAPT treatment, which was not altered in the presence of PACAP 1-38 (Figure 4B,C). *Numb* mRNA expression was not altered significantly after the treatments, such as *TACE* and *CSL* (Figure 4A). Similarly to its protein expression, Numb was elevated during the addition of PACAP 1-38 and DAPT treatment, or with their combined application (Figure 4B,C).

### 2.4. Altered NFATc1 Expression in the Presence of PACAP 1-38 and DAPT

NFATc1 expression was tracked in PACAP-gene-deficient mice, where the mRNA expression of this transcription factor was slightly increased, while a strong elevation was detected in protein levels (Figure 5A–C). In the presence of PACAP 1-38, the mRNA expression of *NFATc1* did not significantly change, but a strong increase in protein expression was observed in the UMR-106 cell line (Figure 5D–F). Moreover, the inhibition of Notch signaling with DAPT reduced the protein expression of the transcription factor, while, during combined application, PACAP 1-38 was able to compensate for the effects of DAPT in vitro (Figure 5D–F). In the femurs of PACAP KO mice, the nuclear localization of NFATc1 was significantly higher than in their WT littermates (Figure 5G,H). In the UMR-106 cell line, PACAP 1-38 induced the elevated expression and nuclear translocation of NFAT1c, but DAPT significantly reduced the nuclear appearance of the transcription factor. On the other hand, during the combined treatment, the presence of PACAP 1-38 compensated the expression-decreasing effect of DAPT and elevated the NFATc1 nuclear translocation (Figure 5G,H).

### 2.5. PACAP 1-38 Alters Notch Signalisation in the Intramembranous Ossification Model

MC3T3-E1 cells are derived from the calvaria of newborn mice (C57BL/6 strain), from the frontal and parietal bones. MC3T3-E1 cells are capable of differentiation into mature osteoblasts and are suitable in vitro models of intramembranous ossification. Therefore, we followed the effects of PACAP 1-38 on the major Notch signaling elements. The Notch receptors’ mRNA expression did not significantly change after PACAP 1-38 administration, similarly to our observations in UMR-106 cells (Figure 6A), but their protein expression significantly decreased in the presence of the neuropeptide (Figure 6B,C). The mRNA expression of *DLL1* and *DLL3* ligands was significantly elevated (Figure 6A); moreover, their protein expression significantly increased, similarly to our observations in the UMR-106 cell line (Figure 6B,C). *DLL4* mRNA and protein expression were barely detectable in the MC3T3-E1 cell line, but an elevation was demonstrated in the presence of PACAP 1-38 (Figure 6A–C). Similarly, the Jagged1 and Jagged2 ligands were detectable in these cells and their mRNA expression was not altered with the addition of PACAP 1-38, but significantly increased protein expression was demonstrated in the presence of the neuropeptide as in the UMR-106 cell line (Figure 6A–C). The most important transcription factor of Notch signaling is CSL, which was detectable in the MC3T3-E1 cell line and did not show alteration in mRNA expression in the PACAP-1-38-administered experimental groups, but its protein expression was significantly elevated by adding PACAP 1-38 (Figure 6A–C). The calcification process was also similar to the UMR-106 cell line, as MC3T3-E1 cells accumulated more Ca^2+^ after the PACAP 1-38 treatment (Figure 6D). Moreover, the nuclear translocation of NFATc1 was elevated in the presence of PACAP 1-38, such as in UMR-106 osteoblastic cells (Figure 6E,F).

## 3. Discussion

Here, we provide experimental proof that PACAP signaling positively acts on the expression of Notch signaling pathways’ elements; this interaction can be proposed as one of the mechanisms by which PACAP enhances osteogenesis in various experimental models.

Such interactions of the two pathways are known in the development of other tissues, but only sparse data are available concerning skeletal tissues. One of the best known PACAP and Notch crosstalk examples is found in the central nervous system, where PACAP and Notch3 activation exerts a common effect on the differentiation of olfactory ensheathing cells [35]. In cerebral ischemia, bioinformatic analysis has demonstrated that Notch signaling elements can be potential target genes involved in PACAP’s protective effects [36]. In tooth development, the signaling crosstalk between Notch and PACAP pathways has been demonstrated [33], but it has not yet been described in skeletal tissues. Understanding whether similar interactions occur in other skeletal tissues, such as cartilage, bone, or periodontal ligament, could provide insights into their development, maintenance, and regenerative processes. Future investigations employing molecular and cellular approaches could elucidate whether the Notch and PACAP pathways intersect in these tissues, potentially unveiling novel targets for regenerative therapies or disease interventions.

Pituitary adenylate cyclase-activating polypeptide (PACAP) has diverse physiological roles, including neuroprotection [37], the modulation of neurotransmitter release [38], and the regulation of circadian rhythms [39]. It is primarily expressed in the central nervous system but is also found in various peripheral tissues [40], including bone [41]. The peptide exerts its effects by binding to specific receptors, mainly PAC1R, which activates adenylate cyclase, leading to increased levels of cAMP and subsequent downstream signaling pathways [40]. Bone development or osteogenesis is a complex process involving the differentiation of mesenchymal stem cells into osteoblasts, the formation of a mineralized extracellular matrix, and the remodeling of bone tissue [42]. Recent studies have illuminated the role of PACAP in this process, suggesting that it may serve as a crucial modulator of bone formation and maintenance [15]. For instance, PACAP has been shown to enhance the expression of osteogenic markers, such as ALP, osteocalcin, and collagen type I [16]. The parallel activation of these genes and the presence of their protein products unambiguously demonstrate the early stages of osteoblast differentiation and the initiation of biomineralization. The mechanism behind this promotion appears to involve the activation of signaling pathways, particularly the cAMP/PKA pathway, which is critical for osteogenesis [41]. In addition to promoting osteoblast differentiation, PACAP plays a significant role in bone mineralization [15]. Studies have demonstrated that the presence of PACAP enhances the mineralization process in osteoblast cultures [16,43]. This effect is likely mediated through the upregulation of matrix proteins and mineral deposition, resulting in stronger and more resilient bone tissue [15,21]. PACAP is emerging as an important regulator of bone development, influencing osteoblast differentiation, mineralization and the balance between bone formation and resorption. Its role in responding to mechanical loading [17] and modulating inflammation [19] further underscores its potential significance in maintaining bone health.

Notch signaling is a highly conserved cell communication pathway that is based on the direct cell–cell connection provided by ligand–receptor interactions. Notch signaling plays a crucial role in various developmental processes, including cell fate determination [44], differentiation [45] and tissue homeostasis [46]. The Notch pathway consists of four Notch receptors (Notch1–4) and five ligands (Jagged1, Jagged2, and Delta-like ligands 1, 3 and 4). When a Notch receptor on one cell binds to a ligand on an adjacent cell, it triggers a series of proteolytic cleavages that release the NICD [34]. TACE plays a critical role in the Notch signaling pathway by facilitating the proteolytic cleavage of Notch receptors. Upon ligand binding, TACE is activated to cleave the extracellular domain of the Notch receptor. This cleavage allows the NICD to be released, which can then translocate to the nucleus and initiate the transcription of target genes [34]. ADAM9 is involved in the proteolytic processing of Notch receptors. This is crucial because Notch signaling requires the cleavage of the Notch receptor to release its intracellular domain (NICD), which then translocate to the nucleus to activate target genes. ADAM9 can facilitate the initial cleavage of Notch, enabling its activation. ADAM9 also affects the availability of Notch ligands, such as Jagged and DLL [34]. CSL is a key transcription factor in the Notch signaling pathway that mediates the effects of Notch activation. When a Notch receptor binds to its ligand, it triggers a series of proteolytic cleavages that release the NICD. The NICD then translocates to the nucleus, where it interacts with CSL to convert it from a repressor to an activator of target gene expression [34]. Numb binds to the Notch receptor and inhibits its ability to interact with the ligand. This interference effectively reduces the activation of Notch signaling in the cell that retains Numb [34]. In the context of bone development, Notch signaling is emerging as a critical regulator, influencing the differentiation of bone cells, the balance between osteogenesis and adipogenesis, and bone remodeling [47]. Notch signaling has been shown to play a significant role in the differentiation of osteoblasts, the cells responsible for bone formation [48]. Studies indicate that activation of Notch signaling in mesenchymal stem cells promotes their differentiation into osteoblasts while inhibiting adipogenic differentiation [49]. This balance is particularly important since excessive adipogenesis can lead to reduced bone density and strength. The Notch pathway influences key osteogenic transcription factors, such as Runx2 [50], which is essential for osteoblast differentiation and function.

Both the PACAP and Notch signaling pathways play pivotal roles in the differentiation of osteoblasts. It has been shown that PACAP has a direct effect on BMP signaling in UMR-106 cells [16] and it also regulates the expression of collagen type I secretion and ALP activity in vivo [15]. Furthermore, the absence of PACAP alters the Ca^2+^ content of long bones and, subsequently, results in increased bone fragility. It is interesting to note that PACAP KO mice showed disturbances in pathological processes such as callus formation [21], but they hardly have any disorders in bone homeostasis under physiological conditions. On the other hand, in in vitro cell cultures, the activation of Notch signaling has been reported to either enhance or suppress the differentiation and mineralization of osteoblasts [31]. Conversely, the inhibition of Notch signaling has also been shown to either facilitate or impede the differentiation and mineralization of osteoblasts [51,52]. Furthermore, sporadic data exist about skeletal disorder of Notch KO mice, but the deletion or mutation of one or more genes within the Notch signaling pathway has been associated with severe skeletal phenotypes in both humans and mouse models [53]. This indicates that a synergistic signalization may be activated in the absence of PACAP or in Notch regulation to compensate for the absence or overproduction of the neuropeptide to maintain proper skeletogenesis. Further evidence has been demonstrated in PACAP-gene-deficient mice, as the expression of Notch signaling elements was elevated and only slight disorders have been found in tooth morphology [33].

In our experiments, we were able to demonstrate the presence of Notch1–3 receptors, but we did not find any expression of the Notch4 receptor in the femurs of adult mice or in UMR-106 osteoblastic cell lines. Additionally, this lack of expression was consistent in the intramembranous ossification model. Notch4 plays a role in vasculogenesis [54] but not in skeletal development, indicating the importance of Notch1–3 receptors in osteogenesis. Interestingly, Notch receptor expression was decreased in femurs and in MC3T3-E1 cells in the absence or presence of PACAP, but elevated expression was demonstrated in the presence of PACAP in UMR-106 cell lines. In UMR-106 cells, the expression pattern of Notch receptors showed differences, which is not surprising, as Notch receptor function and expression timing are crucial in the activation or inhibition of osteogenesis by Notch signaling [52]. These results also indicate the synergistic activation of the two signaling pathways and the PACAP-independent Notch receptor expression regulation in osteogenesis. On the other hand, the augmentation of DLL expression of Notch signaling was demonstrated in the absence or presence of PACAP in vivo and in vitro. Although the mRNA expression levels were not altered, this phenomenon has been shown in chondrogenesis or in osteoblasts [16,17], which suggests the possible posttranscriptional effects of PACAP. When DLLs are elevated, they enhance the activation of Notch receptors, leading to a cascade of intracellular signaling events [55]. This activation can result in several outcomes, including the promotion of cell fate decisions, the inhibition of differentiation into certain lineages, and the maintenance of stem cell characteristics [55]. For example, in the context of hematopoiesis, elevated DLL expression can facilitate the expansion of hematopoietic stem cells by limiting their differentiation [56]. In various organs and tissues, DLL elevation has been implicated in processes such as neurogenesis, where it helps regulate neuronal differentiation by influencing the balance between neural stem cell maintenance and differentiation [57,58]. Interestingly, with the treatment of PACAP 1-38, the expression of these ligands was normalized during Notch signaling inhibition. Altogether, PACAP 1-38 can directly affect DLL function to maintain the normal differentiation processes of osteogenic cell and, thus, regulate proper bone formation.

In Jagged1 and 2 ligands, diverse results were detected, but, in in vitro experiments, the protein expression of Jagged1 was elevated in the presence of PACAP. It has been shown that the activation of Jagged1 does not always show parallelism with Notch receptor expression and can be activated by other signaling pathways [59]. On the other hand, Jagged2 expression does not have a crucial function in skeletogenesis [60]. Subsequently, the downstream targets of Notch signaling, such as CSL, TACE, Adam9 or Numb, showed a strong elevation in the absence or in the presence of PACAP 1-38; moreover, the neuropeptide was able to compensate for the inhibitory effects of DAPT. This also indicates the indirect function of PACAP signalization in Notch activation and well-balanced regulation. Notch regulates both pro- and anti-proliferative effects during development, in stem cells, with its impact varying based on the cellular context [61]. Additionally, it has the capacity to promote proliferation through both cell-autonomous and non-cell-autonomous mechanisms. The involvement of interacting genes and cross-signaling pathways is crucial for modulating the proliferative response to Notch signals [62]. On the other hand, PACAP 1-38 has been proven to have positive effects on UMR-106 cell division [16]. In fact, the combination of PACAP 1-38 and DAPT did not result in higher proliferation, also suggesting the independent but substituent effects of the two signalizations. Moreover, the inhibition of Notch signaling resulted in reduced bone matrix production, including organic and inorganic components, but the addition of PACAP 1-38 was able to compensate for these effects. Indeed, it is likely that PACAP signaling activation can trigger a bypass or a synergistic pathway that substitutes for the direct activation of the Notch receptor, instead promoting ligand expression and further enhancing the downstream targets of the Notch signaling pathway [63].

The crosstalk between NFATc1 (Nuclear Factor of Activated T-cells c1) and Notch signaling is a crucial aspect of many biological processes, particularly in the immune system. NFATc1 and Notch signaling can act synergistically to regulate target gene expression [64]. When both pathways are activated, NFATc1 can enhance the transcriptional activity of NICD by facilitating its binding to specific enhancers or promoters [65]. On the other hand, both NFATc1 and PACAP signaling pathways involve calcium as a critical second messenger. PACAP can stimulate calcium influx through its receptors [66], leading to the activation of signaling pathways that can modulate NFATc1 activity [67]. This interaction can enhance NFATc1-mediated transcription of target genes. Moreover, CSL and NFATc1 form a protein complex and compete for binding to DNA consensus sequences, leading to the reciprocal inhibition of Notch and NFATc1 transactivation [64]. In vitro, NFATc1, similarly to Notch, affects the expression of osteoblast gene markers, indicating that their interaction acts as a local regulatory mechanism governing osteoblast differentiation [68]. In chondrogenic cultures, it has been demonstrated that PACAP increases the activity of PP2B, an upstream element of NFAT activation, and, subsequently, it can induce NFAT translocation to the nuclei of chondrogenic cells [69]. In the presence or absence of PACAP, the NFATc1 nuclear translocation increased and the neuropeptide compensated for the inhibitory effect of Notch signaling blockage. This further indicates that Notch and PACAP signaling can substitute for each other, being responsible for the maintenance of a well-balanced signaling during osteogenesis.

Although we demonstrated possible synergistic crosstalk between PACAP and Notch signaling in bone development, this study has several limitations and further experiments are needed to clarify the exact mechanism. First, many of the data derive from in vitro models, such as UMR-106 and MC3T3-E1 cell lines, which may not fully recapitulate the complex cellular, tissue and systemic interactions occurring in vivo during bone formation and remodeling. Consequently, the extent to which these findings can be translated to physiological or pathological conditions in living organisms remains uncertain. The observation that PACAP deficiency leads to elevated Notch signaling elements and minor skeletal abnormalities suggests compensatory mechanisms [30]. Additionally, our assessment of receptor and ligand expression was limited to mRNA and protein levels at specific time points; dynamic changes over developmental stages or in response to biomechanical or systemic cues are not captured. We also did not explore other signaling pathways—such as BMP, WNT, or inflammatory pathways—that are known to influence osteogenesis and could compensate for or modulate PACAP–Notch interactions [16]. Furthermore, the study lacks functional assays, such as chromatin immunoprecipitation (ChIP), to directly demonstrate transcriptional regulation or physical interactions between PACAP downstream effectors and Notch pathway components, limiting mechanistic clarity [70]. Based on our findings and the current literature, we propose several mechanistic models for the interaction between PACAP and Notch signaling during osteogenesis, such as PACAP, via cAMP/PKA activation, which promotes osteoblast differentiation and mineralization. When PACAP signaling is diminished, Notch pathway components (e.g., Notch1–3, DLLs) can be upregulated as a compensatory response to maintain osteogenic function, suggesting a feedback mechanism that preserves bone formation under stress or deficiency conditions [71]. Cross-modulation via ligand and receptor regulation, such as PACAP, may influence Notch pathway activity indirectly by modulating the expression or stability of Notch ligands (DLLs, Jagged1/2) and receptors. For example, PACAP-induced cAMP signaling could enhance DLL expression or activity post-transcriptionally, thereby activating Notch receptors in a ligand-dependent manner. Conversely, Notch activation may feed back to regulate PACAP receptor expression or signaling efficacy. Substitution and redundancy in signaling, as our data suggest, mean that PACAP can compensate for Notch pathway inhibition, possibly by activating alternative transcriptional programs or promoting the ligand-independent activation of Notch downstream effectors [72]. This indicates a potential redundancy where PACAP signaling sustains osteogenic gene expression when Notch activity is compromised. Future studies employing functional assays such as ChIP, co-immunoprecipitation, and live-cell imaging of pathway dynamics, especially at different developmental stages, are needed to validate these models [73]. Investigating the role of other intersecting pathways will further clarify the complexity of osteogenic regulation and help delineate the precise mechanistic interplay between PACAP and Notch signaling during bone development. Modulating the PACAP pathway and the Notch signaling pathways offers potential therapeutic avenues for bone-related disorders such as osteoporosis, fractures and developmental bone diseases. The coordinated targeting of these pathways might optimize bone regeneration, balancing osteoblast proliferation and differentiation.

Nonetheless, we would like to emphasize that this is the first study that provides data that prove the synergistic connection of Notch and PACAP signalization in bone development. As these two pathways converge, they can co-regulate target genes that are critical for osteogenesis. Moreover, the interplay between Notch and PACAP signaling can result in a more finely tuned regulatory response, allowing cells and tissues to better adapt to environmental signals. This type of interaction may lead to enhanced osteoblast differentiation, the increased expression of osteogenic markers, and improved tissue resilience. Therefore, the coordinated action of Notch and PACAP signaling pathways not only promotes effective cellular communication but also enhances overall developmental outcomes.

## 4. Materials and Methods

### 4.1. Animals

The generation and maintenance of PACAP-deficient mice on the CD1 background have been detailed previously [74,75]. These mice underwent backcrossing with the CD1 strain for at least ten generations. Genotyping was confirmed through PCR techniques. For the experiments, we euthanized four-month-old wild-type (WT, n = 10) and homozygous PACAP-deficient (PACAP KO, n = 10) female and male mice. The animals were provided with food and water ad libitum and were maintained under a 12 h light/dark cycle. All procedures were conducted following the ethical guidelines approved by the University of Pécs (permission number: BA02/2000-15024/2011).

### 4.2. Cell Culture

The rat osteosarcoma cell line UMR-106 (ATCC^®^ CRL-1661™) was used to study osteogenic differentiation as a model of long bone osteoblasts [76]. The MC3T3-E1 cell line (ATCC^®^ CRL-1661™) derived from mouse calvaria was used to model intramembranous ossification [77]. Cells were maintained in high-glucose Dulbecco’s Modified Eagle’s Medium (DMEM) (PAA Laboratories, Pasching, Austria), enriched with 10% fetal bovine serum (FBS) (PAA Laboratories) at 37 °C with 5% CO_2_ and 80% humidity in a CO_2_ incubator. Upon reaching 70% confluence of either UMR-106 or MC3T3-E1 cells, the medium was replaced with DMEM lacking FBS to initiate osteogenic differentiation, marked as day 0. For UMR-106 passage numbers 46, 49, 52 and 55 and the MC3T3 E1 cell line, 16, 19 and 25 passage numbers were used in at least three independent experiments. The differentiation of cells occurred for one week, and treatments were applied for four days, as previously described [16].

### 4.3. PACAP 1-38 and DAPT Treatments

PACAP 1-38 was administered at a concentration of 100 nM (stock solution: 100 μM, dissolved in sterile distilled water) to act as a PAC1 receptor agonist. As a Notch signaling inhibitor, DAPT (N-[N-(3,5-Difluorophenacetyl-L-alanyl)]-S-phenylglycine t-butyl ester) (Sigma-Aldrich, St. Louis, MO, USA) was continuously applied at 5 μM (stock solution: 5 mM, dissolved in sterile distilled water) starting on day 1.

### 4.4. Determination of Non-Toxic Concentrations of DAPT and PACAP

To evaluate the non-toxic concentration of DAPT in UMR-106 cells, 1 µM, 5 µM, and 10 µM of DAPT were added to the cell cultures after differentiation for four days. To assess viability and mitochondrial activity, 25 μL of MTT reagent (3-[4,5-dimethylthiazolyl-2]-2,5-diphenyltetrazolium bromide; 25 mg MTT/5 mL PBS) (Sigma-Aldrich, St. Louis, MO, USA) was added to each Petri dish on day 4. Following a 2 h incubation at 37 °C and the addition of 500 μL of MTT solubilizing solution, absorbance was measured at 570 nm (Chameleon, Hidex Ltd., Turku, Finland). It has been shown in MC3T3-E1 cells that DAPT successfully inhibits Notch signaling [78,79]; therefore, we attempted to use the same concentration of the inhibitor. The effective concentration of PACAP was determined earlier, as demonstrated in our previous study [16,69].

### 4.5. Staining for Light Microscopy

UMR cells from various experimental groups were cultured on round coverslips (Menzel-Gläser, Menzel GmbH, Braunschweig, Germany) placed in Petri dishes (PAA Laboratories). On day 4, cells were fixed using a 4:1 mixture of absolute ethanol and 40% formaldehyde. Alizarin Red (Sigma-Aldrich, St. Louis, MO, USA) staining for calcium-rich deposits and the von Kossa method (Millipore, Billerica, MA, USA) were used to demonstrate calcium phosphate presence in cell cultures. Staining protocols were performed according to the manufacturer’s instructions. Photomicrographs were captured using an Olympus Bx53 microscope with constant camera and exposure settings (Olympus, Tokyo, Japan).

### 4.6. Proliferation and Mitochondrial Activity Assessment

On day 4, DMEM containing 1 μCi/mL ^3^H-thymidine (diluted from Thymidine [6-^3^H] 20–30 Ci/mmol; 0.74–1.11 TBq/mmol, American Radiolabeled Chemicals, Inc., St. Louis, MO, USA) was added to the cultures for 16 h. Cells were subsequently fixed with ice-cold 5% trichloroacetic solution for 20 min and then collected into opaque 96-well microtiter plates (Wallac, PerkinElmer Life and Analytical Sciences, Shelton, CT, USA). After air-drying for one week, radioactivity was measured using a Chameleon liquid scintillation counter (Chameleon, Hidex Ltd., Turku, Finland). To evaluate general cell viability or mitochondrial activity, 25 μL of MTT reagent (3-[4,5-dimethylthiazolyl-2]-2,5-diphenyltetrazolium bromide; 25 mg MTT/5 mL PBS) (Sigma-Aldrich, St. Louis, MO, USA) was added to each Petri dish on day 4. Following a 2 h incubation at 37 °C and the addition of 500 μL of MTT solubilizing solution, absorbance was read at 570 nm (Chameleon, Hidex Ltd., Turku, Finland).

### 4.7. RT-PCR Analysis

The CG-200 Freezer/Mill was used as a cryogenic mill that precools in liquid nitrogen and grinds bone samples (Cole-Parmer, Vermon Hills, IL, USA). Samples were ground for 30 sec till they became a fine powder then were dissolved in Trizol (Applied Biosystems, Foster City, CA, USA). After the addition of 20% RNase-free, chloroform samples were centrifuged at 4 °C at 10,000× *g* for 15 min. Samples were incubated in 500 µL of RNase-free isopropanol at −20 °C for 1 h, then total RNA was harvested in RNase-free water and stored at −20 °C.

Cells from the UMR-106 and MC3T3-E1 cell lines were lysed in Trizol (Applied Biosystems, Foster City, CA, USA). Following the addition of 20% RNase-free chloroform, samples were centrifuged at 4 °C and 10,000× *g* for 15 min. RNA was precipitated in 500 µL of RNase-free isopropanol at −20 °C for 1 h, then resuspended in RNase-free water and stored at −20 °C. The reverse transcriptase reaction mixture consisted of 2 µg RNA, 0.112 µM oligo(dT), 0.5 mM dNTP and 200 units of high-capacity RT (Applied Biosystems) in 1 × RT buffer. For primer sequences and additional details about the polymerase chain reactions, refer to Table 1. Amplifications were conducted in a Labnet MultiGene™ 96-well Gradient Thermal Cycler (Labnet International, Edison, NJ, USA) over a final volume of 21 μL (including 1 μL forward and reverse primers [0.4 μM], 0.5 μL dNTP [200 μM] and 5 units of Promega GoTaq^®^ DNA polymerase in 1× reaction buffer) under the following conditions: 95 °C for 2 min, followed by 35 cycles (denaturation at 94 °C for 1 min; annealing at specified temperatures for 1 min; extension at 72 °C for 90 s), culminating with a final extension at 72 °C for 10 min. The PCR products were separated via electrophoresis on a 1.2% agarose gel containing ethidium bromide, using Actin as an internal control. The optical density of signals was quantified using ImageJ 1.40g software, and the results were normalized against the optical density of WT or untreated control cultures.

### 4.8. Western Blot Analysis

Tissues were cryo-ground in CG-200 Freezer/Mill (Cole-Parmer, Vermon Hills, IL, USA); strong mechanical destruction may result in protein fragmentation. After centrifugation, tissue pellets were suspended in 100 μL of homogenization RIPA. Samples were stored at −70 °C. Suspensions were sonicated by pulsing burst for 30 s at 40 A (Cole-Parmer, Vermon Hills, IL, USA). For Western blotting, total tissue lysates were used.

Following a wash with physiological saline, UMR-106 and MC3T3-E1 cells were harvested and centrifuged. The cell pellets were resuspended in 100 μL of RIPA buffer (comprising 150 mM sodium chloride, 1.0% NP40, 0.5% sodium deoxycholate and 50 mM Tris at pH 8.0) containing protease inhibitors (Aprotinin [10 μg/mL], 5 mM Benzamidine, Leupeptin [10 μg/mL], Trypsin inhibitor [10 μg/mL], 1 mM PMSF, 5 mM EDTA, 1 mM EGTA, 8 mM Na-Fluoride, 1 mM Na-orthovanadate). Samples were stored at −70 °C. The suspensions were sonicated for 30 s pulses at 40 A (Cole-Parmer, Vermon Hills, IL, USA). Bone samples and cell populations can present unique challenges due to their composition and the presence of various mineralized matrices, which may affect protein extraction efficiency and overall signal quality. The weak signals or saturation can be the result of strong mechanical degradation, which can hardly be compensated for in bone tissue. A BCA protein assay was used to determine the protein concentration of each individual sample to ensure equal loading by diluting each sample with Laemmli solution to a concentration of 2 µg/µL. Therefore, total cell lysates were prepared with Laemmli electrophoresis sample buffer (4% SDS, 10% 2-mercaptoethanol, 20% glycerol, 0.004% bromophenol blue, 0.125 M Tris HCl at pH 6.8) and boiled for 10 min.

Consequently, 20 µL of the protein solution, containing 40 µg of protein, was loaded in each experiment and samples were separated by 7.5% SDS-PAGE to detect proteins such as Notch1, Notch2, Notch3, DLL1, DLL3, DLL4, Jagged1, Jagged2, CSL, TACE Numb, Adam9, NFATc1, PKA, Collagen I, ALP and Actin. Following electrophoresis, proteins were transferred to nitrocellulose membranes. After blocking with 5% non-fat dry milk in phosphate-buffered saline (PBST) containing 0.1% Tween 20, membranes were exposed to primary antibodies overnight at 4 °C, at the dilutions specified in Table 2. Following a 30 min wash in PBST, membranes were incubated with anti-rabbit IgG (Bio-Rad Laboratories, Hercules, CA, USA) at a dilution of 1:1500, anti-goat IgG (Sigma) at 1:2000, and anti-mouse IgG (Bio-Rad Laboratories) at 1:1500. Signals were detected using enhanced chemiluminescence (Advansta, San Jose, CA, USA) according to the instructions of the manufacturer. Signals were developed in the FluorochemE gel documentary system (Protein Simple, San Jose, CA, USA). The optical density of Western blot signals was assessed using ImageJ 1.40g software; only an approximate percentage was given to indicate the direction of the change. The initial results were normalized using multiple loading controls, such as GAPDH and Actin. The data were then normalized relative to WT samples and untreated control cultures. We conducted a densitometry analysis of Western blots, presented as mean values with standard deviations.

### 4.9. Immunocytochemistry

Bone tissue samples were washed in PBS (phosphate buffer solution) three times and fixed in 10% formalin fixative for 72 h. Bones were decalcified in 4% EDTA (Sigma-Aldrich, St. Louis, MO, USA) for four weeks until the bones became soft. Afterwards, the decalcifying solution was washed out with PBS for 30 min and the samples were embedded in paraffin. Serial sections of 7 µm thick slides were prepared with a rotation microtome (Leica, Wetzlar, Germany). After deparaffinization in descending alcohol raw and washing in PBST (phosphate-buffered saline supplemented with 1% Tween-20) three times, unspecific binding sites were blocked with bovine serum albumin (BSA) (Amresco, Solon, OH, USA) at 37 °C 30 min; then, the slides were washed in PBS 3 × 10 min in PBS.

On day 4, immunocytochemistry was conducted on UMR-106 and MC3T3-E1 cells cultured on coverslips to visualize the intracellular localization of NFATc1. Cells were fixed in 10% formalin fixative and rinsed in 70% ethanol. After additional washing in PBS (pH 7.4), nonspecific binding sites were blocked using PBST with 1% bovine serum albumin (Amresco LLC, Solon, OH, USA). Cultures and bones were then exposed to polyclonal anti-NFATc1 antibody (Abcam, Cambridge, UK) at a dilution of 1:500 and incubated at 4 °C overnight. The primary antibody was visualized using anti-mouse Alexa555 secondary antibody (Life Technologies Corporation, Carlsbad, CA, USA) at a dilution of 1:1000. The specificity of the antibody was confirmed by applying control peptide identical to the antigen for which the antibody was raised, yielding no nonspecific signals. Cultures and tissue slides were mounted in a Vectashield mounting medium (Vector Laboratories, Peterborough, England) containing DAPI for nuclear DNA staining. For the negative control, samples without a primary antibody were used (Appendix A); for the positive control, chondrocytes were used as demonstrated earlier [80]. For the subcellular localization of NFATc1, fluorescent images were captured with an Olympus FV3000 confocal microscope (Olympus Co., Tokyo, Japan) utilizing a 60× oil immersion objective (NA: 1.3). The excitation laser line was set at 543 nm, with an average pixel time of 4 µs. The Z-image series with 1 µm optical thickness were recorded in sequential scan mode. The images of Alexa555 and DAPI were subsequently overlaid with Adobe Photoshop version 10.0. The red pixel intensity in the nucleus was measured using ImageJ software with RGB measurement settings in at least 10 cells in 3 independent experiments. The initial results were normalized to a blue background; then, the data were normalized relative to the WT samples and untreated control cultures. The nuclear intensity of NFATc1 is presented as mean values with standard deviations.

### 4.10. ALP Activity Assay

Cells were washed in physiological NaCl solution and were harvested. After centrifugation, cell pellets were suspended in 100 μL of homogenization RIPA-buffer containing the protease inhibitors mentioned above. Suspensions were sonicated using a pulsing burst for 3 × 10 sec at 40 A (Cole-Parmer, Vermon Hills, IL, USA) on ice. After centrifugation at 10,000× *g* for 10 min at 4 °C, supernatants of the samples were used for in vitro enzyme activity measurements. Untreated cultures were used as controls. An alkaline phosphatase assay kit (Colorimetric) (ab83369 Abcam) was used following the manufacturer’s instructions. Results were measured on 420 nm (Chameleon, Hidex Ltd., Turku, Finland). The activity of ALP was calculated according to the manufacturer’s protocol.

### 4.11. Statistical Analysis

All data are representative of at least three different experiments. Where applicable, data are expressed as the mean ± SEM. Statistical analysis was performed using one-way ANOVA tests combined with post hoc tests in UMR-106 cells and Students’ *t* test in PACAP KO mice and MC3T3E1 cells. Where ANOVA reported significant differences among the groups (*p* < 0.05) a post hoc test (multiple comparison versus control group, Dunnett’s method) was used to isolate the groups that differed from the control group at *p* < 0.05. The respective control group was the untreated control when comparison was made among control, PACAP-1-38-treated and DAPT-treated and DAPT+PACAP1-38 groups, whereas the PACAP-1-38-treated cultures were used as the control in the post hoc test when a comparison was made among the DAPT-treated and DAPT+PACAP-1-38-treated samples. The same statistical analysis was used during the Western blot densitometry and quantification of the NFATc1 nuclear presence.

## Figures and Tables

**Figure 1 ijms-26-05088-f001:**
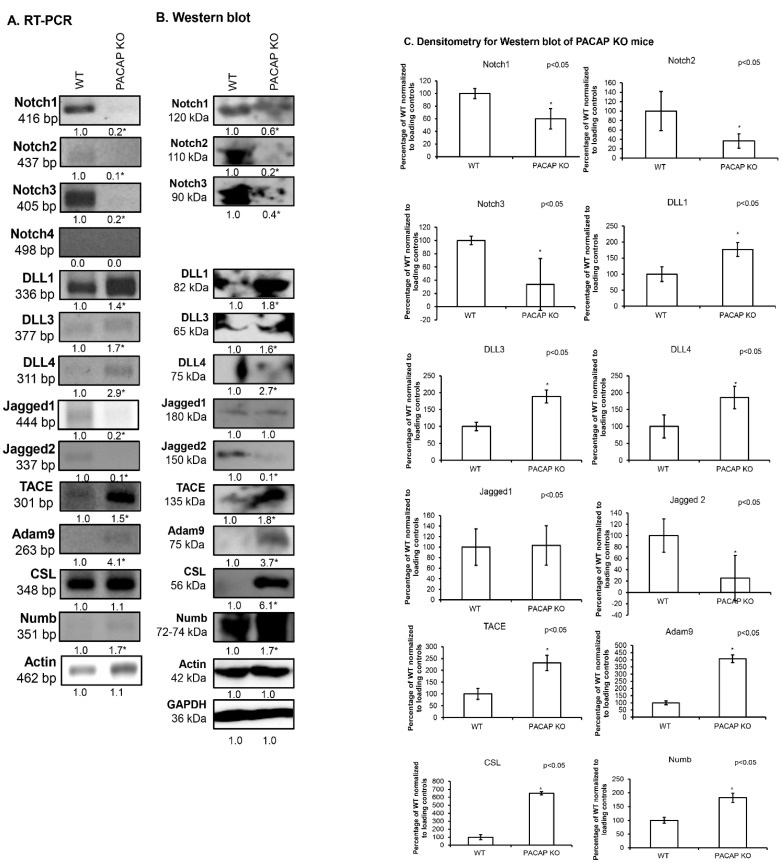
Expression of Notch signaling in femurs of PACAP KO mice. mRNA (RT-PCR) (**A**) and protein (Western blot) (**B**) expression of Notch1, 2, 3, 4, DLL1, 3, 4, Jagged1, 2, TACE, Adam9, CSL and Numb. For RT-PCR reactions, *Actin*, and for Western blot reactions, Actin and GAPDH were used as loading controls. The optical density of signals was measured and the results were normalized to the optical density of the WT mice samples. Representative data of three independent experiments are shown. For panels (**A**,**B**), numbers below signals represent the integrated densities of signals determined using ImageJ 1.40g freeware software and normalized to Actin (RT-PCR) or Actin and GAPDH (Western blot). Densitometry and statistical analysis of Western blot results of PACAP KO mice (**C**). All data are the average of at least three different experiments. All data were normalized to GAPDH and Actin. Data are expressed as the mean ± SEM. Asterisks indicate significant (* *p* < 0.05) alterations in PACAP KO as compared to WT.

**Figure 2 ijms-26-05088-f002:**
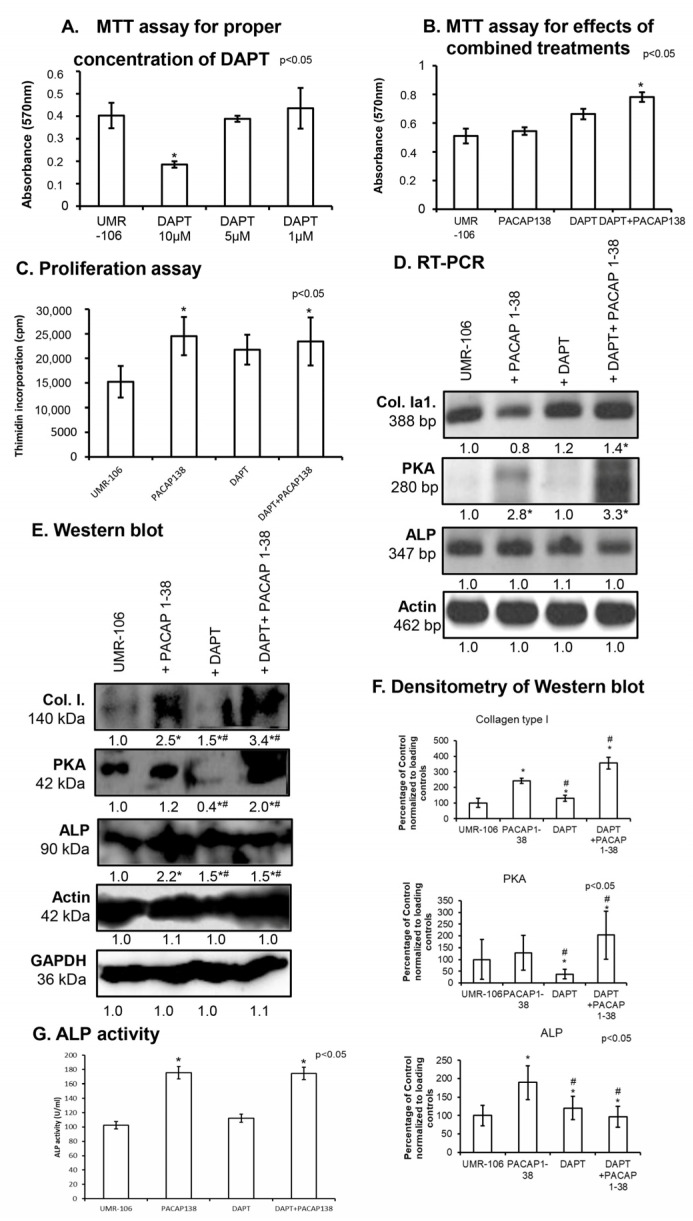
Effects of DAPT on the UMR-106 cell line. Effects of administration of 1 µM, 5 µM, and 10 µM DAPT on mitochondrial metabolic activity (MTT assay) (**A**) in the UMR-106 cell line on culturing day four. Effects of PACAP 1-38, DAPT, and combined treatment on the UMR-106 cell line. Effects of PACAP 1-38, DAPT, and DAPT+PACAP1-38 administration on mitochondrial metabolic activity (MTT assay) (**B**) and cellular proliferation (**C**) in the UMR-106 cell line on culturing day four. Asterisks indicate significant (* *p* < 0.05) alterations in mitochondrial activity and cell proliferation as compared to the respective control. mRNA (RT-PCR) (**D**) and protein (Western blot) (**E**) expression of collagen type I, PKA, and ALP. For RT-PCR *Actin* and for Western blot reactions, Actin and GAPDH were used as loading controls. The optical density of signals was measured, and the results were normalized to the optical density of *Actin* (RT-PCR) and Actin and GAPDH (Western blot) and compared to control cultures. For panels (**C**,**D**), numbers below signals represent the integrated densities of signals, determined using ImageJ software. Asterisks indicate significant (* *p* < 0.05) alterations compared to the respective control, and ^#^ indicates a significant difference (^#^
*p* < 0.05) compared to the PACAP-1-38-treated group. Densitometry and statistical analysis of Western blot results of UMR-106 cells (**F**). All data are the average of at least three different experiments. Statistical analysis was performed using one-way ANOVA tests combined with post hoc tests. All data were normalized to GAPDH and Actin. Data are expressed as the mean ± SEM. Asterisks indicate significant (* *p* < 0.05) alterations compared to the respective control, ^#^ indicates a significant difference (^#^
*p* < 0.05) compared to PACAP-1-38-treated group. (**G**) Enzyme activity of ALP in UMR-106 cells on day 4. Asterisks indicate a significant (* *p* < 0.05) increase in activity as compared to the respective control. Representative data of three independent experiments.

**Figure 3 ijms-26-05088-f003:**
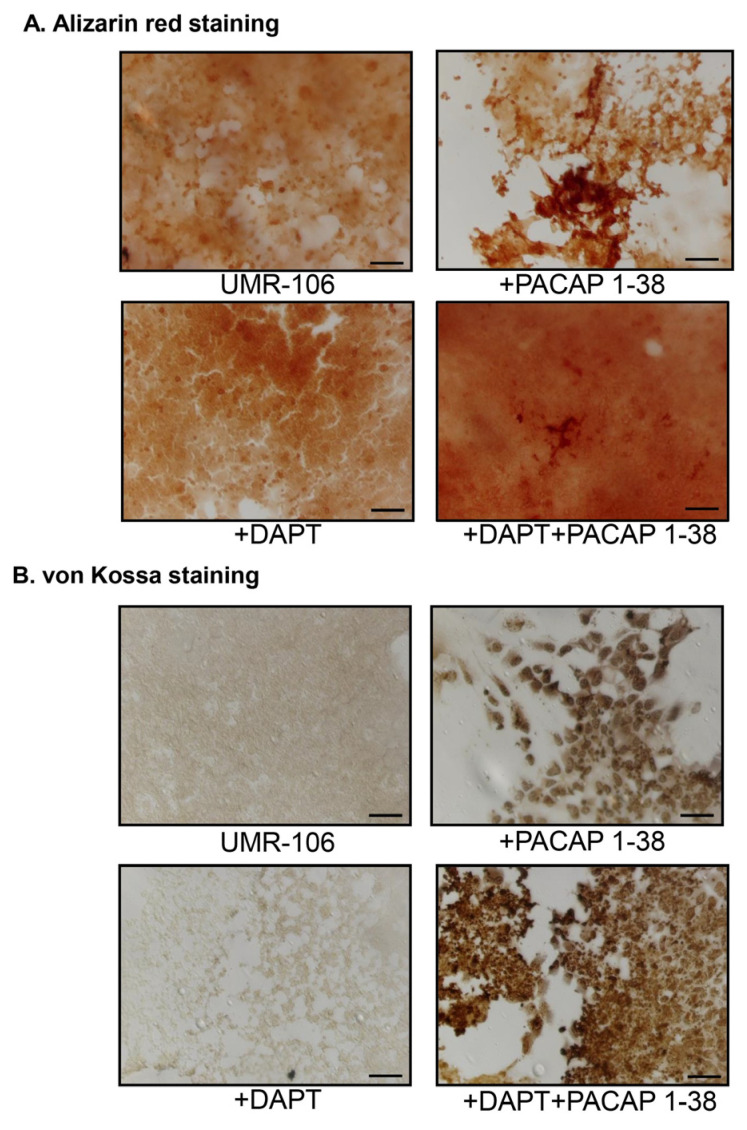
Effects of PACAP 1-38 and Notch inhibition on the Ca^2+^ accumulation of UMR-106 cells. (**A**) Effects of PACAP 1-38, DAPT, and DAPT+PACAP1-38 administration. Extracellular Ca^2+^-deposits of 4-day-old UMR-106 cells were visualized using Alizarin red staining. Original magnification was 20×. Scale bar, 100 µm. (**B**) Extracellular Ca^2+^-phosphate crystals were detected using the von Kossa method on day four of culturing. Original magnification was 20×. Scale bar, 100 µm.

**Figure 4 ijms-26-05088-f004:**
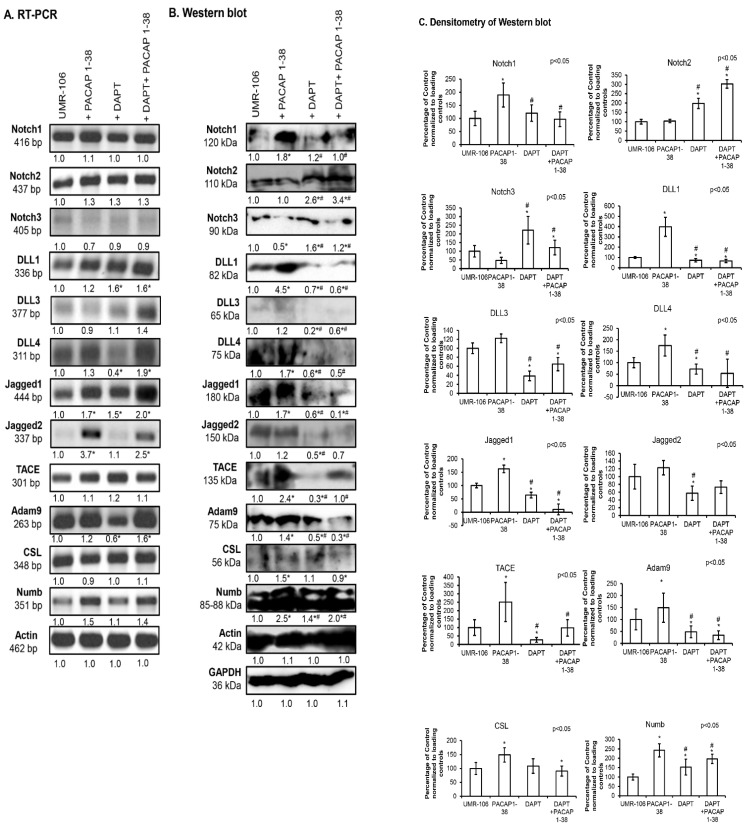
Expression of Notch signaling after PACAP 1-38, DAPT and combined treatment in the UMR-106 cell line. mRNA (RT-PCR) (**A**) and protein (Western blot) (**B**) expression of Notch1, 2, 3, DLL1, 3, 4, Jagged1, 2, TACE, Adam9, CSL and Numb. For RT-PCR *Actin* and for Western blot reactions, Actin and GAPDH were used as loading controls. The optical density of signals was measured, and the results were normalized to the optical density of *Actin* (RT-PCR) and Actin and GAPDH (Western blot) and compared to control cultures. Representative data of three independent experiments are given. For panels (**A**,**B**), numbers below signals represent the integrated densities of signals determined using ImageJ software. Densitometry and statistical analysis of Western blot results of UMR-106 cells (**C**). All data are the average of at least three different experiments. Statistical analysis was performed using one-way ANOVA tests combined with post hoc tests. All data were normalized on GAPDH and Actin. Data are expressed as the mean ± SEM. Asterisks indicate significant (* *p* < 0.05) alterations compared to the respective control, and ^#^ indicates a significant difference (^#^
*p* < 0.05) compared to the PACAP-1-38-treated group.

**Figure 5 ijms-26-05088-f005:**
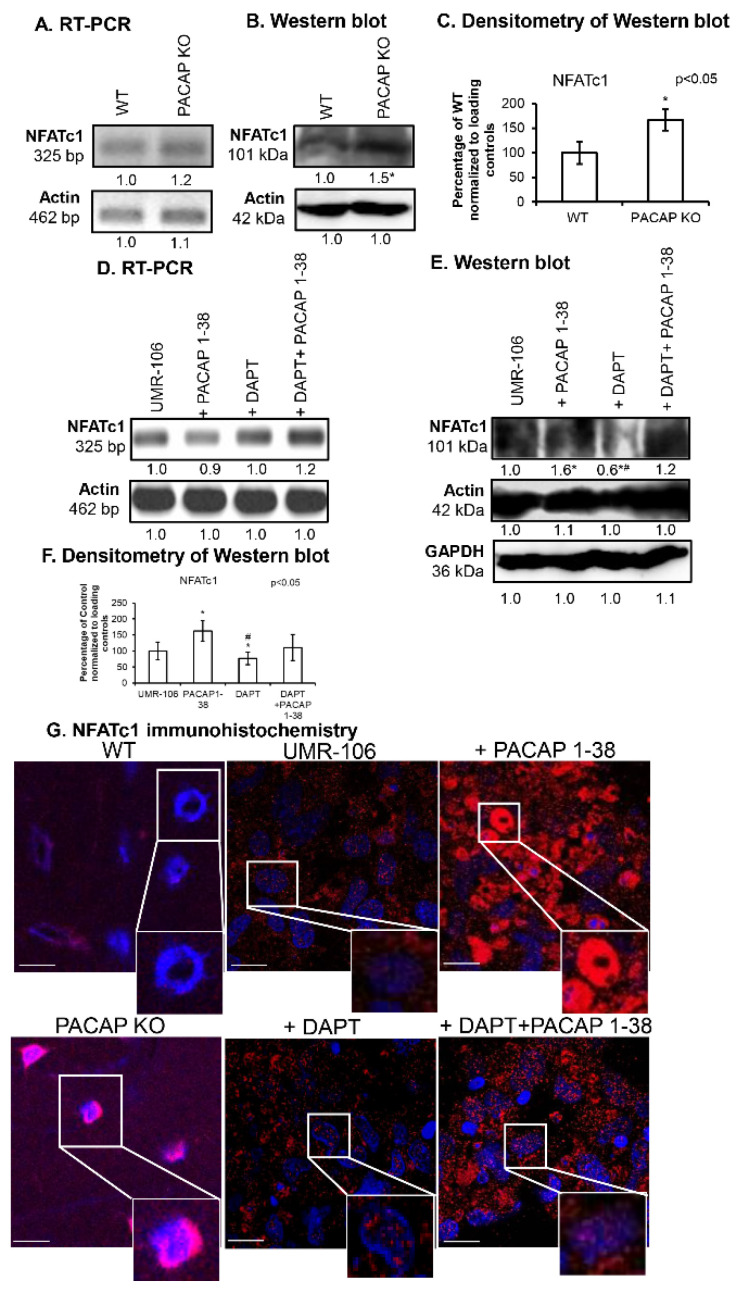
Expression of NFATc1 transcription factor. mRNA (RT-PCR) (**A**) and protein (Western blot) (**B**) expression of NFATc1 in WT and PACAP KO mice. Densitometry and statistical analysis of Western blot results of PACAP KO mice (**C**). All data are the average of at least three different experiments. Statistical analysis was performed using one-way ANOVA tests combined with post hoc tests. All data were normalized to GAPDH and Actin. Data are expressed as the mean ± SEM. mRNA (RT-PCR) (**D**) and protein (Western blot) (**E**) expression of NFATc1 in UMR-106 cell line after PACAP 1-38, DAPT and DAPT+PACAP 1-38 treatments. For RT-PCR Actin and for Western blot reactions, Actin and GAPDH were used as loading controls. The optical density of signals was measured, and results were normalized to the optical density of *Actin* (RT-PCR) and Actin and GAPDH (Western blot) and compared to control cultures. For panels (**A**–**D**), numbers below signals represent the integrated densities of signals determined using ImageJ software. Densitometry and statistical analysis of Western blot results of UMR-106 cells (**F**). All data are the average of at least three different experiments. Statistical analysis was performed by one-way ANOVA tests combined with post hoc tests. All data were normalized to GAPDH and Actin. Data are expressed as mean ± SEM. Asterisks indicate significant (* *p* < 0.05) alteration of PACAP KO as compared to WT and PACAP 1-38, DAPT, DAPT+PACAP 1-38 as compared to the representative control. ^#^ indicates a significant difference (^#^
*p* < 0.05) compared to the PACAP-1-38-treated group. (**G**) Immunocytochemistry of NFATc1 in femurs of WT and PACAP KO mice and in UMR-106 cells after PACAP 1-38, DAPT and DAPT+PACAP 1-38 treatments on day 4 of culturing. Original magnification was 60×. Scale bar, 10 µm. Representative results of three independent experiments are shown. Quantification of NFATc1 nuclear translocation (**H**). Nuclear intensity of NFATc1, presented as mean values with standard deviations. All data are the average of at least three different experiments. Statistical analysis was performed using one-way ANOVA tests combined with post hoc tests. Data are expressed as the mean ± SEM. Asterisks indicate significant (* *p* < 0.05) alteration of PACAP KO as compared to WT and PACAP 1-38, DAPT, DAPT+PACAP 1-38 as compared to representative control. ^#^ indicates a significant difference (^#^
*p* < 0.05) compared to the PACAP-1-38-treated group.

**Figure 6 ijms-26-05088-f006:**
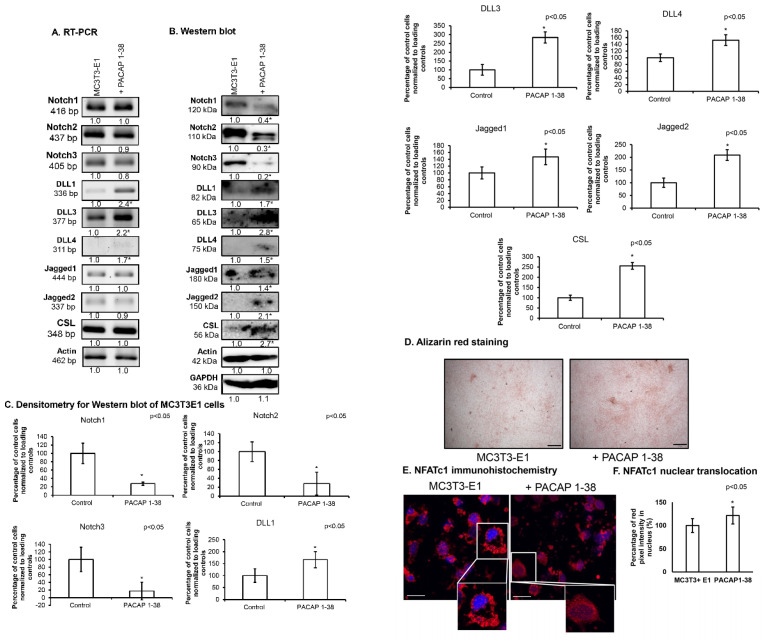
Expression of Notch signaling after PACAP 1-38 treatment in the MC3T3-E1 cell line. mRNA (RT-PCR) (**A**) and protein (Western blot) (**B**) expression of Notch1, 2, 3, DLL1, 3, 4, Jagged1, 2 and CSL. For RT-PCR *Actin* and for Western blot reactions, Actin and GAPDH were used as controls. Optical density of signals was measured, and the results were normalized to the optical density of control cultures. For panels (**A**,**B**), numbers below signals represent integrated densities of signals determined using ImageJ software. Densitometry and statistical analysis of Western blot results of MC3T3-E1 cells (**C**). All data are the average of at least three different experiments. Statistical analysis was performed using one-way ANOVA tests combined with post hoc tests. All data were normalized to GAPDH and Actin. Data are expressed as the mean ± SEM. Asterisks (**A**–**C**) indicate significant (* *p* < 0.05) alterations in the PACAP 1-38 treatment group as compared to the control. (**D**) Effects of PACAP 1-38 administration on extracellular Ca^2+^ deposits of four-day-old MC3T3-E1 cells with Alizarin red staining. The original magnification was 20×. Scale bar, 100 µm. (**E**) Immunocytochemistry of NFATc1 in MC3T3-E1 cells after PACAP 1-38 treatment on day four of culturing. The original magnification was 60×. Scale bar, 10 µm. Representative data of three independent experiments. Quantification of NFATc1 nuclear translocation (**F**). Nuclear intensity of NFATc1, presented as mean values with standard deviations. All data are the average of at least three different experiments. Data are expressed as the mean ± SEM. Asterisks indicate significant (* *p* < 0.05) alterations with PACAP 1-38 as compared to the representative control.

**Table 1 ijms-26-05088-t001:** Nucleotide sequences, amplification sites, GenBank accession numbers, amplimer sizes and PCR reaction conditions for each primer pair are shown.

Gene	Primer	Nucleotide Sequence (5′→3′)	GenBank ID	Annealing Temperature	Amplimer Size (bp)
**Alkaline phosphatase** **(Alpl)**	sense	GAA GTC CGT GGG CAT CGT (474–491)	**NM_013059**	59 °C	347
	antisense	CAG TGC GGT TCC AGA CAT AG (801–820)			
**DLL1** **(Dll1)**	sense	GAA ACA CCA GCC TCC ACC T (2305–2323)	**NM_001379042.1**	53 °C	336
	antisense	GGA ATC TCC CCA CCC CTA (2623–2640)			
**DLL3** **(Dll3)**	sense	CCT GGT TTC CAA GGC TCT AA (1060–1079)	**NM_007866.2**	57 °C	377
	antisense	ACA GCG AAC TCG CAT CTC A (1418–1436)			
**DLL4** **(Dll4)**	sense	TTG CCC TTC AAT TTC ACC (673–690)	**NM_019454.4**	54 °C	311
	antisense	CAG GAC AGG CTG CCA TCT (966–983)			
**Jagged2** **(Jag2)**	sense	CCT CGT CGT CAT TCC CTT TC (395–411)	**NM_001409685.1**	57 °C	337
	antisense	GCA TTC TTT GCC CAT CCA G (713–731)			
**Jagged1** **(Jag1)**	sense	TCA GGC ATG ATA AAC CCT AGC (735–755)	**NM_013822.5**	56 °C	444
	antisense	GGG CTG ATG AGT CCC ACA (1161–1178)			
**Collagen type I** **(Col1a1)**	sense	GGG CGA GTG CTG TGC TTT (348–365)	**NM_007742.3**	60 °C	388
	antisense	GGG ACC CAT TGG ACC TGA A (717–735)			
**Notch1** **(notch1)**	sense	GGA TCA CAT GGA CCG ATT G (6504–6522)	**NM_008714.3**	56 °C	416
	antisense	TGG ATG GAG ACT GCT GGA A (6901–6919)			
**Notch2** **(notch2)**	sense	GTA TCT CCA AGC CGT GTA TG (2791–2810)	**NM_010928.2**	55 °C	437
	antisense	GCA GAA GGG ACC AGT GAA (3210–3227)			
**Notch3** **(notch3)**	sense	GCA CCA GTG ATG GAA TAG GC (2308–2327)	**NM_008716.3**	56 °C	405
	antisense	AGC GAG GAC CAG CAA AGC (2695–2712)			
**Notch4** **(notch4)**	sense	GCC ACT CTT TAG CCA ACG C (3207–3225)	**NM_010929.2**	57 °C	498
	antisense	CAT CGC AGG TCC CAT CAC (3687–3704)			
**NFATc1** **(Nfatc1)**	sense	CCT GAC CAC CGA TAG CAC (973–990)	**NM_001164109.1**	52 °C	325
	antisense	CTC GTA TGG ACC AGA ATG T (1279–1297)			
**Adam9** **(Adam9)**	sense	TGA TTC GCT TAG CAA ACT (857–874)	**NM_001270996.1**	49 °C	263
	antisense	GTG GCT CCT TGA ACA TAC (1102–1119)			
**PKA** **(Prkaca)**	sense	GCA AAG GCT ACA ACA AGG C (847–865)	**NM_008854**	53 °C	280
	antisense	ATG GCA ATC CAG TCA ATC G (1109–1126)			
**CSL** **(Csl)**	sense	TGG AGC TTC CTG GAC AAT (1054–1071)	**NM_027945.4**	51 °C	348
	antisense	AGG CTG GTG GAG TAA ATG (1384–1401)			
**Numb** **(Numb)**	sense	ATT CCG TGT CAC AAC TGC (597–614)	**NM_001136075.3**	51 °C	351
	antisense	AAA TCG GTC TTC CTC TGC (930–947)			
**TACE** **(Adam17)**	sense	AAG TCT GCC TGG CTC ATC (1192–1209)	**NM_001277266.1**	51 °C	301
	antisense	CCT CCT TGG TCC TCA TTT (1475–1492)			
**Actin** **(Actb)**	sense	GCC AAC CGT GAA AAG ATG A (419–437)	**NM_001014970**	48 °C	462
	antisense	CAA GAA GGA AGG CTG GAA AA (861–880)			

**Table 2 ijms-26-05088-t002:** Antibodies used in the experiments.

Antibody	Host Animal	Dilution	Distributor
**Anti-Notch1**	rabbit, polyclonal	1:500	Cell Signaling, Danvers, MA, USA
**Anti-Notch2**	rabbit, polyclonal	1:500	Cell Signaling, Danvers, MA, USA
**Anti-Notch3**	rabbit, polyclonal	1:500	Cell Signaling, Danvers, MA, USA
**Anti-Coll. I.**	mouse, monoclonal	1:1000	Sigma-Aldrich, St. Louis, MO, USA
**Anti-DLL1**	rabbit, polyclonal	1:500	Cell Signaling, Danvers, MA, USA
**Anti-DLL3**	rabbit, polyclonal	1:500	Cell Signaling, Danvers, MA, USA
**Anti-DLL4**	rabbit, polyclonal	1:500	Cell Signaling, Danvers, MA, USA
**Anti-Jagged1**	rabbit, polyclonal	1:500	Cell Signaling, Danvers, MA, USA
**Anti-Jagged2**	rabbit, polyclonal	1:500	Abcam, Cambridge, UK
**Anti-ALP**	rabbit, polyclonal	1:500	Abcam, Cambridge, UK
**Anti-CSL**	rabbit, polyclonal	1:500	Cell Signaling, Danvers, MA, USA
**Anti-TACE**	rabbit, polyclonal	1:500	Cell Signaling, Danvers, MA, USA
**Anti-PKA**	rabbit, polyclonal	1:800	Cell Signaling, Danvers, MA, USA
**Anti-Numb**	rabbit, polyclonal	1:500	Cell Signaling, Danvers, MA, USA
**Anti-Adam9**	rabbit, polyclonal	1:600	Cell Signaling, Danvers, MA, USA
**Anti-NFATc1**	mouse, monoclonal	1:500	Abcam, Cambridge, UK
**Anti-Actin**	mouse, monoclonal	1:10,000	Sigma-Aldrich, St. Louis, MO, USA
**Anti-GAPDH**	rabbit, polyclonal	1:800	Abcam, Cambridge, UK

## Data Availability

Data are contained within the article and Appendix A.

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
