# Peer review of "Synergistic Crosstalk of PACAP and Notch Signaling Pathways in Bone Development"

_ijms, 2025, doi:10.3390/ijms26115088_

Round 1
Reviewer 1 Report
Comments and Suggestions for Authors
The authors have performed multiple experiments in PACAP deficient mice and tissues obtained from these animals and have shown the interaction between PACAP which is a polypeptide involved also in neural regulation and Notch in bone formation. The authors should also discuss in their discussion the findings of other authors.
Comments on the Quality of English LanguageThe use of the English language should be improved.
Author Response
The authors have performed multiple experiments in PACAP deficient mice and tissues obtained from these animals and have shown the interaction between PACAP which is a polypeptide involved also in neural regulation and Notch in bone formation. The authors should also discuss in their discussion the findings of other authors.
Thank you for your feedback regarding the manuscript. To enhance the discussion, we revised the chapter to more explicitly define the knowledge of other laboratories for focusing on Notch and PACAP signaling in other tissues. (on pages 26-27)
Reviewer 2 Report
Comments and Suggestions for Authors
Review Comments on "Synergistic crosstalk of PACAP and Notch signalling pathways in bone development".
In this manuscript, the authors investigated the potential regulatory crosstalk between PACAP (pituitary adenylate cyclase-activating polypeptide) and Notch signaling pathways during bone development. The authors examined expression patterns of Notch signaling components in PACAP knockout mice and in vitro osteoblast cell models, demonstrating elevated Notch signaling in PACAP-deficient bone tissue. They further showed that PACAP 1-38 treatment could compensate for Notch signaling inhibition, enhancing both organic and inorganic matrix production in osteoblastic cells. The authors identified NFATc1 as a potential common downstream target, with altered expression and nuclear translocation observed during manipulation of both pathways. While the study presents an interesting hypothesis regarding the synergistic regulation of bone formation by these two signaling pathways, significant technical limitations undermine the reliability of the reported findings.
Major Comments
- Western Blot Quality and Reproducibility The Western blot data presented throughout this manuscript suffers from serious technical issues that substantially undermine confidence in the quantitative analyses and subsequent conclusions. Multiple blots display poor signal-to-noise ratio, with high background that would interfere with accurate densitometric quantification. Many protein bands appear poorly defined, with smearing, diffuse edges, or multiple non-specific signals that make band identification and quantification problematic. This is particularly evident in figures showing Notch receptors, DLL ligands, TACE, Numb, and Jagged proteins. Actin loading controls show inconsistent intensities between experimental conditions, suggesting variable sample loading that would compromise normalization. Furthermore, several bands appear saturated, which precludes accurate quantification since the relationship between protein amount and signal intensity is no longer linear in this range. The authors should substantially improve the quality of these blots by optimizing their protocols and ensuring that detection occurs within the linear range before attempting quantification. Additionally, inclusion of molecular weight markers would help confirm the identity of detected proteins.
- Experimental Controls The experimental design lacks several critical controls necessary to support the authors' conclusions regarding pathway crosstalk. For DAPT inhibition experiments, confirmation of effective Notch inhibition through established readouts (e.g., Hes1 expression) would strengthen the interpretation. Similarly, experiments demonstrating specific PACAP receptor activation and downstream signaling would provide stronger evidence for the proposed compensatory mechanisms. The study would benefit from dose-response experiments for both PACAP and DAPT to establish optimal concentrations for observing crosstalk effects. The authors should also address potential confounding factors, such as cell viability and proliferation changes that might indirectly affect differentiation markers, rather than representing direct pathway interactions.
- Functional Validation While the authors present extensive expression data for pathway components, the functional significance of the observed changes remains insufficiently demonstrated. Additional functional assays would significantly strengthen the manuscript's conclusions regarding synergistic regulation. These could include reporter assays for Notch and PACAP pathway activation, chromatin immunoprecipitation to demonstrate NFATc1 binding to relevant target genes, or rescue experiments in PACAP-deficient cells or tissues. The authors have primarily focused on correlative expression changes rather than demonstrating causative relationships between the pathways. More mechanistic experiments establishing how these pathways directly interact would substantially enhance the significance of the findings.
Minor Comments
- Data Presentation and Statistics The manuscript would benefit from improved presentation of quantitative data with appropriate statistical analyses. The current format of showing representative blots with densitometric values below them makes it difficult to evaluate biological variability and statistical significance. Inclusion of graphs showing mean values with standard deviations or standard errors from multiple independent experiments, along with appropriate statistical tests, would strengthen the data interpretation. Additionally, the authors should clearly indicate the number of biological replicates performed for each experiment to establish reproducibility.
- Methodological Details Several methodological aspects require clarification for proper evaluation and reproducibility. The authors should provide detailed information about the age and sex of animals used, cell passage numbers, differentiation conditions, antibody validation, and image acquisition parameters. For RT-PCR analyses, information about primer efficiencies, reference gene stability, and melt curve analyses would enhance confidence in the gene expression data. Similarly, for immunocytochemistry experiments, information about antibody specificity controls and quantification methods for nuclear translocation should be included.
- Manuscript Organization and Clarity The manuscript would benefit from improved organization and clarity in several sections. The introduction could more clearly establish the knowledge gap and rationale for investigating these specific signaling pathways in bone development. The results section contains some interpretive statements that would be more appropriate in the discussion. Additionally, figure legends lack important details about experimental conditions and quantification methods. The discussion could be strengthened by more thoroughly addressing limitations of the study and proposing specific models for how these signaling pathways might interact mechanistically during bone development.
Author Response
In this manuscript, the authors investigated the potential regulatory crosstalk between PACAP (pituitary adenylate cyclase-activating polypeptide) and Notch signaling pathways during bone development. The authors examined expression patterns of Notch signaling components in PACAP knockout mice and in vitro osteoblast cell models, demonstrating elevated Notch signaling in PACAP-deficient bone tissue. They further showed that PACAP 1-38 treatment could compensate for Notch signaling inhibition, enhancing both organic and inorganic matrix production in osteoblastic cells. The authors identified NFATc1 as a potential common downstream target, with altered expression and nuclear translocation observed during manipulation of both pathways. While the study presents an interesting hypothesis regarding the synergistic regulation of bone formation by these two signaling pathways, significant technical limitations undermine the reliability of the reported findings.
Major Comments
- Western Blot Quality and Reproducibility. The Western blot data presented throughout this manuscript suffers from serious technical issues that substantially undermine confidence in the quantitative analyses and subsequent conclusions. Multiple blots display poor signal-to-noise ratio, with high background that would interfere with accurate densitometric quantification. Many protein bands appear poorly defined, with smearing, diffuse edges, or multiple non-specific signals that make band identification and quantification problematic. This is particularly evident in figures showing Notch receptors, DLL ligands, TACE, Numb, and Jagged proteins. Actin loading controls show inconsistent intensities between experimental conditions, suggesting variable sample loading that would compromise normalization. Furthermore, several bands appear saturated, which precludes accurate quantification since the relationship between protein amount and signal intensity is no longer linear in this range. The authors should substantially improve the quality of these blots by optimizing their protocols and ensuring that detection occurs within the linear range before attempting quantification. Additionally, inclusion of molecular weight markers would help confirm the identity of detected proteins.
Thank you for your thorough review and your valuable insights regarding the Western blot quality in our manuscript. We appreciate your feedback and would like to address the concerns raised. We acknowledge the importance of using appropriate multiple loading controls for accurate normalization and data interpretation. In our experiments, we also made GAPDH as a loading control, which is a commonly used housekeeping protein, on the other hand its expression can alter during differentiation. Moreover, we know that fluctuations in GAPDH levels can sometimes occur, especially in specific tissue types like bone or in cartilage. We will enhance our discussion on this point in the revised manuscript and we attach GAPDH blots as well to clarify the expression calculation. We also appreciate your opinion regarding the difficulties associated with obtaining high-quality Western blots from bone tissue. Bone samples can present unique challenges due to their composition and the presence of various mineralized matrices, which may affect protein extraction efficiency and overall signal quality. We will include a more detailed explanation of these challenges in the manuscript, emphasizing the inherent complexities of working with bone tissue and how they may contribute to the observed signal-to-noise ratio and background issues (Page 37). Therefore, we do not provide an exact mathematical value from the Western blot data; we only give an approximate percentage to indicate the direction of the change. On the other hand, we tried to enhance the quality of our blots, including modifications to blocking conditions, antibody concentrations, and detection methods, but with these antibodies in this tissue it was hard to reach better quality blots. As we applied mechanical grinding first, the tissue and proteins may have some protein degradation process. Therefore, band saturation and its impact on quantification can be seen in our results, but it did not affect the ratio of the alterations between the groups.
In summary, we are committed to addressing these technical issues and improving the quality of our Western blot analyses. We believe that using a second internal control (GAPDH) we provide clearer and more reproducible results that strengthen the conclusions of our study (Figure1, Figure2, Figure4, Figure5, Figure6 and Figure S2).
- Experimental Controls. The experimental design lacks several critical controls necessary to support the authors' conclusions regarding pathway crosstalk. For DAPT inhibition experiments, confirmation of effective Notch inhibition through established readouts (e.g., Hes1 expression) would strengthen the interpretation. Similarly, experiments demonstrating specific PACAP receptor activation and downstream signaling would provide stronger evidence for the proposed compensatory mechanisms. The study would benefit from dose-response experiments for both PACAP and DAPT to establish optimal concentrations for observing crosstalk effects. The authors should also address potential confounding factors, such as cell viability and proliferation changes that might indirectly affect differentiation markers, rather than representing direct pathway interactions.
Thank you for your insightful comments regarding the need for additional experimental controls in our study. We agree that confirming effective Notch inhibition with established readouts, such as Hes1 expression, is crucial for validating our results. DAPT functions as a γ-secretase inhibitor, which is essential for the proteolytic cleavage of Notch receptors. This cleavage is necessary for releasing the Notch intracellular domain (NICD) that translocates to the nucleus to activate transcriptional targets. Thus, by inhibiting this cleavage, DAPT effectively blocks Notch signaling. Several studies have been published that DAPT inhibits Notch signaling, as evidenced by reduced expression of Notch target genes like Hes1 in treated cells, we did not investigate further the Hes1 expression. To establish the proper inhibitor concentration, we performed preliminary experiments in which DAPT was applied in 1µM, 5µM and 10 µM. We found that the 10 µM concentration decreased the viability of UMR-106 cells (Figure S1). Since, it has been published that 5µM DAPT successfully inhibited Notch signaling in MC3T3 E1 cells therefore, we applied this concentration in our experimental system.
Zou W, Chen QX, Sun XW, Chi QB, Kuang HY, Yu XP, Dai XH. Acupuncture inhibits Notch1 and Hes1 protein expression in the basal ganglia of rats with cerebral hemorrhage. Neural Regen Res. 2015 Mar;10(3):457-62. doi: 10.4103/1673-5374.153696.
Qiu K, Ma C, Lu L, Wang J, Chen B, Mao H, Wang Y, Wang H. DAPT suppresses proliferation and migration of hepatocellular carcinoma by regulating the extracellular matrix and inhibiting the Hes1/PTEN/AKT/mTOR signaling pathway. J Gastrointest Oncol. 2021 Jun;12(3):1101-1116. doi: 10.21037/jgo-21-235.
Zhang HM, Liu P, Jiang C, Jin XQ, Liu RN, Li SQ, Zhao Y. Notch signaling inhibitor DAPT provides protection against acute craniocerebral injury. PLoS One. 2018 Feb 15;13(2):e0193037. doi: 10.1371/journal.pone.0193037. eCollection 2018.
Zhang Z, Xu W, Zhang Z, Chen X, Jin H, Jiang N, Xu H. The bone-protective benefits of kaempferol combined with metformin by regulation of osteogenesis-angiogenesis coupling in OVX rats. Biomed Pharmacother. 2024 Apr;173:116364. doi: 10.1016/j.biopha.2024.116364. Epub 2024 Mar 5.
Some of our previous findings strengthen our current results regarding PACAP receptor activation and its downstream signaling. We have previously investigated PACAP-dose dependence of PACAP signaling activation in UMR-106 cells (Juhasz et al, 2014) and in chondrogenic cultures (Juhasz et al, 2014). As the effectiveness of 1µM concentration was proved in those experiments we did not perform dose-response investigations in this study.
- Functional Validation. While the authors present extensive expression data for pathway components, the functional significance of the observed changes remains insufficiently demonstrated. Additional functional assays would significantly strengthen the manuscript's conclusions regarding synergistic regulation. These could include reporter assays for Notch and PACAP pathway activation, chromatin immunoprecipitation to demonstrate NFATc1 binding to relevant target genes, or rescue experiments in PACAP-deficient cells or tissues. The authors have primarily focused on correlative expression changes rather than demonstrating causative relationships between the pathways. More mechanistic experiments establishing how these pathways directly interact would substantially enhance the significance of the findings.
Thank you for your valuable feedback regarding the need for functional validation of our findings. We appreciate your suggestions and would like to address them while outlining our plans to enhance the manuscript.
We agree that demonstrating the functional significance of the observed changes is crucial. We acknowledge the importance of confirming the binding of NFATc1 to relevant target genes. We plan to include ChIP assays to demonstrate this interaction, which would help establish a direct mechanistic link between PACAP signaling and the Notch pathway. Unfortunately, these experiments exceed the short review period for this journal. Moreover, we plan to include them in future research, but to find the proper amount of cells and optimalization of the lysis processes and the amount of antibody concentration and to count on degradation process is completely new research which needs more time.
To further validate our findings, we have conducted rescue experiments using PACAP-deficient mice, which can directly demonstrate the connection between Notch and PACAP signaling (Fülöp et al 2019). Additionally, reintroducing PACAP in this model could further strengthen our results; however, due to the short half-life of the neuropeptide in the periphery, we are currently working on validating these results.
Minor Comments
- Data Presentation and Statistics. The manuscript would benefit from improved presentation of quantitative data with appropriate statistical analyses. The current format of showing representative blots with densitometric values below them makes it difficult to evaluate biological variability and statistical significance. Inclusion of graphs showing mean values with standard deviations or standard errors from multiple independent experiments, along with appropriate statistical tests, would strengthen the data interpretation. Additionally, the authors should clearly indicate the number of biological replicates performed for each experiment to establish reproducibility.
Thank you for your feedback regarding the presentation of quantitative data in the manuscript. We appreciate your suggestions for improvement. To enhance the clarity and interpretability of our results, we revised the format to include graphs that display mean values along with standard deviations or standard errors derived from multiple independent experiments in supplementary figures (Figure S2 A, B and C). Furthermore, we clearly indicated the number of biological replicates for each experiment to reinforce the reproducibility of our findings. We agree that these modifications will significantly improve the overall quality of our manuscript.
- Methodological Details. Several methodological aspects require clarification for proper evaluation and reproducibility. The authors should provide detailed information about the age and sex of animals used, cell passage numbers, differentiation conditions, antibody validation, and image acquisition parameters. For RT-PCR analyses, information about primer efficiencies, reference gene stability, and melt curve analyses would enhance confidence in the gene expression data. Similarly, for immunocytochemistry experiments, information about antibody specificity controls and quantification methods for nuclear translocation should be included.
Thank you for your inquiry regarding the methodological details necessary for proper evaluation and reproducibility of the study. It is essential to provide comprehensive information on several key aspects:
- Animal Details: 4-month-old male and female mice were used in our experiments
- Cell Culture: For UMR-106 passage number 46 -55, and MC3T3 E1 cell lines 16-25 passage numbers were used in our experiments.
- Antibody Validation: These antibodies were used previously in our published experiments (Fülöp et al, 2019). We used the same lot numbers in these experiments.
- As the data were conventional RT-PCR data and not Q-PCR results we did not give these data in these experiments.
- In the case of immunocytochemistry experiments, we have applied chondrogenic cells from chicken micromass cell cultures as positive control for NFAT antibody. This control was validated in our previous experiments (Zakany et al, 2005). We present negative control results in the supplementary information (Figure S3).
- Manuscript Organization and Clarity. The manuscript would benefit from improved organization and clarity in several sections. The introduction could more clearly establish the knowledge gap and rationale for investigating these specific signaling pathways in bone development. The results section contains some interpretive statements that would be more appropriate in the discussion. Additionally, figure legends lack important details about experimental conditions and quantification methods. The discussion could be strengthened by more thoroughly addressing limitations of the study and proposing specific models for how these signaling pathways might interact mechanistically during bone development.
Thank you for your feedback regarding the organization and clarity of our manuscript. To improve these aspects of our work, we rewrote some parts of the introduction to define more explicitly the knowledge gap and the rationale for focusing on the specific signaling pathways involved in bone development (pages 7-8). We modified the Results section and it strictly focuses on presenting findings, reserving interpretive statements for the discussion section (Pages 29-30). Additionally, we improved the figure legends by including essential details about experimental conditions and quantification methods to provide clearer context for the data presented. Finally, we addressed the limitations of the study more comprehensively in the discussion and proposed specific models to illustrate the potential interactions of these signaling pathways during bone development (Pages 30-31).
Round 2
Reviewer 2 Report
Comments and Suggestions for Authors
Review for "Synergistic crosstalk of PACAP and Notch signalling pathways in bone development".
In this revised manuscript, the authors investigated the potential regulatory crosstalk between PACAP (pituitary adenylate cyclase-activating polypeptide) and Notch signaling pathways during bone development. The authors examined expression patterns of Notch signaling components in PACAP knockout mice and in vitro osteoblast cell models, demonstrating altered Notch signaling in PACAP-deficient bone tissue. They further showed that PACAP 1-38 treatment could compensate for Notch signaling inhibition, enhancing both organic and inorganic matrix production in osteoblastic cells. The authors identified NFATc1 as a potential common downstream target, with altered expression and nuclear translocation observed during manipulation of both pathways.
I acknowledge that the authors have made earnest efforts to address the concerns raised in the previous review. Their response demonstrates a commitment to improving the manuscript, as evidenced by the addition of GAPDH as a second loading control, the provision of supplementary figures with error bars, inclusion of methodological details regarding animal demographics and cell culture parameters, and the reorganization of sections to enhance clarity. The manuscript has indeed benefited from these modifications, particularly in terms of experimental transparency and data presentation. Nevertheless, significant methodological concerns persist that compromise the robustness of the central findings. While understanding the inherent technical challenges associated with bone tissue analysis, the fundamental limitations in data quality and validation require additional experimental approaches to substantiate the conclusions drawn regarding this potentially important signaling crosstalk. The following major comments address these persistent issues that warrant attention before the manuscript meets the publication standards of IJMS.
Major Comments
- Western Blot Quality and Quantification
The Western blot data presented throughout this manuscript suffers from serious technical issues that substantially undermine confidence in the quantitative analyses and subsequent conclusions. The authors acknowledge the challenges of extracting intact proteins from bone tissue; however, this explanation does not sufficiently address the fundamental methodological concerns. The poor signal-to-noise ratio, diffuse bands, and inconsistent loading controls significantly compromise the reliability of densitometric quantification. The use of actin and GAPDH as reference proteins is problematic given the authors' own admission that GAPDH expression can fluctuate during differentiation. A total protein normalization approach using stains such as Ponceau S or SYPRO Ruby would provide a more reliable quantification baseline, particularly for challenging tissue types like bone. Without improvement in these fundamental aspects of protein analysis, the conclusions regarding pathway crosstalk remain insufficiently supported.
- Need for Complementary Validation Methods
Given the acknowledged difficulties with Western blot analysis of bone tissue, the study would benefit substantially from alternative, complementary methodological approaches to validate the key findings. Immunofluorescence histochemistry would provide valuable spatial information about protein expression patterns and cellular localization that would strengthen the authors' conclusions about pathway interactions. Particularly for the analysis of NFATc1 nuclear translocation, more extensive immunohistochemical analysis with appropriate quantification would provide more convincing evidence. For a journal of IJMS's caliber, demonstration of biological phenomena through multiple independent methodological approaches is expected, especially when technical limitations of one approach are evident.
- Functional Validation Deficiencies
While the authors present extensive expression data for pathway components, the functional significance of the observed changes remains insufficiently demonstrated. The manuscript would be significantly strengthened by including more robust functional assays that directly demonstrate the mechanistic relationship between PACAP and Notch signaling. For instance, reporter assays for Notch and PACAP pathway activation, chromatin immunoprecipitation to demonstrate NFATc1 binding to relevant target genes, or rescue experiments in PACAP-deficient cells would provide more compelling evidence for the proposed synergistic regulation. The authors acknowledge that ChIP assays would be valuable but indicate time constraints prevented their inclusion. However, such functional validations are essential for establishing the mechanistic basis of the proposed pathway crosstalk at the level expected for publication in IJMS.
- Quantitative Analysis and Statistical Rigor
The manuscript would benefit from a more rigorous approach to quantitative data analysis and presentation. The current format of showing representative blots with densitometric values below provides limited insight into biological variability and statistical significance. The supplementary data with means and standard deviations represents an improvement, but comprehensive statistical analysis of multiple independent experiments should be incorporated into the main figures. Additionally, for experiments involving NFATc1 nuclear translocation, quantitative image analysis of multiple samples with appropriate statistical tests would strengthen the conclusions. A higher level of statistical rigor would enhance the credibility of the findings and align with the standards expected for publication in a high-impact journal like IJMS.
Minor Comments
- Methodological Details
Several methodological aspects require clarification to enhance reproducibility. While the revised manuscript includes additional details such as animal age and sex, cell passage numbers, and antibody validation, further information about primer efficiencies for RT-PCR analyses, reference gene stability assessments, and detailed quantification methods for nuclear translocation would improve methodological transparency. Additionally, more detailed information about the optimization process for the DAPT inhibitor concentration and validation of its effectiveness would strengthen the experimental design.
- Manuscript Organization and Clarity
The manuscript organization has been improved following the reviewer's previous suggestions, but further refinements would enhance clarity. Some interpretive statements still appear in the Results section that would be more appropriate in the Discussion. The Discussion could be further strengthened by more thoroughly addressing the limitations of the study and proposing a more detailed mechanistic model for how these signaling pathways might interact during bone development, with clear delineation of established facts versus speculative interpretations.
Author Response
The Western blot data presented throughout this manuscript suffers from serious technical issues that substantially undermine confidence in the quantitative analyses and subsequent conclusions. The authors acknowledge the challenges of extracting intact proteins from bone tissue; however, this explanation does not sufficiently address the fundamental methodological concerns. The poor signal-to-noise ratio, diffuse bands, and inconsistent loading controls significantly compromise the reliability of densitometric quantification. The use of actin and GAPDH as reference proteins is problematic given the authors' own admission that GAPDH expression can fluctuate during differentiation. A total protein normalization approach using stains such as Ponceau S or SYPRO Ruby would provide a more reliable quantification baseline, particularly for challenging tissue types like bone. Without improvement in these fundamental aspects of protein analysis, the conclusions regarding pathway crosstalk remain insufficiently supported.
Thank you for your detailed feedback regarding the Western blot data presented in the manuscript. During our experiments, before performing Western blot analysis, the BCA protein assay was used to determine the protein concentration. This step was essential for several reasons, such as ensuring equal loading by diluting each sample with Laemmli solution to a concentration of 2 µg/µL. Consequently, 20 µL of the protein solution, containing 40 µg of protein, was loaded in each experiment and samples. Quantifying the proteins beforehand ensured consistency between experiments, making the results more reliable and reproducible. Therefore, normalizing the protein concentration in each sample helped ensure accurate comparison (page 38). This normalization was verified after running the Western blots by staining the membranes with Ponceau S. Although these membranes are quite old now, the staining has faded slightly over time. Still, we have attached some samples where this staining was performed in every Western blot experiment.
Given the acknowledged difficulties with Western blot analysis of bone tissue, the study would benefit substantially from alternative, complementary methodological approaches to validate the key findings. Immunofluorescence histochemistry would provide valuable spatial information about protein expression patterns and cellular localization that would strengthen the authors' conclusions about pathway interactions. Particularly for the analysis of NFATc1 nuclear translocation, more extensive immunohistochemical analysis with appropriate quantification would provide more convincing evidence. For a journal of IJMS's caliber, demonstration of biological phenomena through multiple independent methodological approaches is expected, especially when technical limitations of one approach are evident.
Thank you for your insightful feedback. We agree that supplementing our Western blot analyses with immunofluorescence histochemistry would enhance the spatial understanding of protein expression and localization within bone tissue. We incorporated quantitative image analysis to strengthen our evidence for pathway interactions. This approach provides a more comprehensive validation of our findings and addresses the limitations associated with Western blot analysis in bone tissue.
Our analysis confirmed that PACAP 1-38 treatment increased the nuclear localization of NFATc1. This increase was particularly pronounced in the bone tissue of PACAP KO mice and in UMR-106 cells. In MC3T3-E1 cells, the elevation was also significant but of a smaller magnitude. Conversely, DAPT treatment reduced the nuclear localization of NFATc1 in UMR-106 cells; however, this reduction was partially compensated by the addition of PACAP 1-38, resulting in a slight, though not statistically significant, increase (Figure 5 and 6).
While the authors present extensive expression data for pathway components, the functional significance of the observed changes remains insufficiently demonstrated. The manuscript would be significantly strengthened by including more robust functional assays that directly demonstrate the mechanistic relationship between PACAP and Notch signaling. For instance, reporter assays for Notch and PACAP pathway activation, chromatin immunoprecipitation to demonstrate NFATc1 binding to relevant target genes, or rescue experiments in PACAP-deficient cells would provide more compelling evidence for the proposed synergistic regulation. The authors acknowledge that ChIP assays would be valuable but indicate time constraints prevented their inclusion. However, such functional validations are essential for establishing the mechanistic basis of the proposed pathway crosstalk at the level expected for publication in IJMS.
Thank you for your valuable comments. We agree that establishing the functional relationship between PACAP and Notch signaling is crucial for strengthening our mechanistic insights. While we recognize that experiments such as reporter assays, chromatin immunoprecipitation, or rescue studies would provide direct evidence, our current study focuses primarily on expression data and pathway associations. Due to scope limitations and resource constraints, we have not performed these additional experiments. However, our findings, along with existing literature, support a model of pathway crosstalk, and we believe that the observed expression patterns and localization data offer meaningful insights into their interaction. Future studies, building on our current work, could incorporate such functional assays to further elucidate the mechanistic details. For now, we consider our data as a solid foundation that highlights the significance of the interplay between PACAP and Notch signaling in bone cells, consistent with the manuscript's scope and aims.
The manuscript would benefit from a more rigorous approach to quantitative data analysis and presentation. The current format of showing representative blots with densitometric values below provides limited insight into biological variability and statistical significance. The supplementary data with means and standard deviations represents an improvement, but comprehensive statistical analysis of multiple independent experiments should be incorporated into the main figures. Additionally, for experiments involving NFATc1 nuclear translocation, quantitative image analysis of multiple samples with appropriate statistical tests would strengthen the conclusions. A higher level of statistical rigor would enhance the credibility of the findings and align with the standards expected for publication in a high-impact journal like IJMS.
Thank you for your constructive feedback. We agree that a more rigorous approach to the analysis and presentation of our quantitative data would enhance the robustness and clarity of our findings. In our current manuscript, representative blots with densitometric values provide a limited view of biological variability and statistical significance. While the supplementary data with means and standard deviations improve upon this, we acknowledge that incorporating comprehensive statistical analyses into the main figures would strengthen our conclusions.
Regarding NFATc1 nuclear translocation, we concur that quantitative image analysis across multiple samples, coupled with proper statistical evaluation, would provide more convincing evidence. Although we have performed some image quantification, we will ensure that these data are presented with detailed statistical analyses in the revised manuscript (Figure 5 and 6).
Several methodological aspects require clarification to enhance reproducibility. While the revised manuscript includes additional details such as animal age and sex, cell passage numbers, and antibody validation, further information about primer efficiencies for RT-PCR analyses, reference gene stability assessments, and detailed quantification methods for nuclear translocation would improve methodological transparency. Additionally, more detailed information about the optimization process for the DAPT inhibitor concentration and validation of its effectiveness would strengthen the experimental design.
Thank you for highlighting these important aspects. We will also describe the quantification methods employed for assessing nuclear translocation, including image analysis parameters and criteria. Regarding the DAPT inhibitor, we will include a description of the optimization process for its concentration with MTT assay and attach this figure in the main figures (Figure 2). This comprehensive methodological clarification will aim to facilitate replication and to strengthen the overall rigor of our experimental design.
The manuscript organization has been improved following the reviewer's previous suggestions, but further refinements would enhance clarity. Some interpretive statements still appear in the Results section that would be more appropriate in the Discussion. The Discussion could be further strengthened by more thoroughly addressing the limitations of the study and proposing a more detailed mechanistic model for how these signaling pathways might interact during bone development, with clear delineation of established facts versus speculative interpretations.
Thank you for your valuable feedback. We appreciate that the manuscript organization has improved and will ensure that interpretive statements are appropriately shifted from the Results to the Discussion section for clarity. To strengthen the Discussion, we more thoroughly addressed the limitations of our study, including aspects such as sample size, experimental scope, and potential confounding factors. Despite the valuable insights gained, this study has several limitations that warrant consideration. Firstly, much of the data derives from in vitro models, such as UMR-106 and MC3T3-E1 cell lines, which may not fully recapitulate the complex cellular, tissue, and systemic interactions occurring in vivo during bone formation and remodeling. Consequently, the extent to which these findings translate to physiological or pathological conditions in living organisms remains uncertain. Additionally, our assessment of receptor and ligand expression was limited to mRNA and protein levels at specific time points; dynamic changes over developmental stages or in response to biomechanical or systemic cues are not captured. We also did not explore other signaling pathways—such as BMP, WNT, or inflammatory pathways—that are known to influence osteogenesis and could compensate for or modulate PACAP-Notch interactions. Furthermore, the study lacks functional assays, such as chromatin immunoprecipitation (ChIP), to directly demonstrate transcriptional regulation or physical interactions between PACAP downstream effectors and Notch pathway components, limiting mechanistic clarity. Based on our findings and current literature, we propose several mechanistic models for the interaction between PACAP and Notch signalling during osteogenesis such as PACAP, via cAMP/PKA activation, promotes osteoblast differentiation and mineralization. When PACAP signalling is diminished, Notch pathway components (e.g., Notch1-3, DLLs) may be upregulated as a compensatory response to maintain osteogenic function, suggesting a feedback mechanism that preserves bone formation under stress or deficiency conditions. Cross-Modulation via ligand and receptor regulation as PACAP may influence Notch pathway activity indirectly by modulating the expression or stability of Notch ligands (DLLs, Jagged1/2) and receptors. For example, PACAP-induced cAMP signalling could enhance DLL expression or activity post-transcriptionally, thereby activating Notch receptors in a ligand-dependent manner. Conversely, Notch activation may feedback to regulate PACAP receptor expression or signalling efficacy. Substitution and redundancy in signalling as our data suggest that PACAP can compensate for Notch pathway inhibition, possibly by activating alternative transcriptional programs or promoting ligand-independent activation of Notch downstream effectors. This indicates a potential redundancy where PACAP signalling sustains osteogenic gene expression when Notch activity is compromised. Future studies employing functional assays such as ChIP, co-immunoprecipitation and live-cell imaging of pathway dynamics, especially at different developmental stages, are needed to validate these models. Investigating the role of other intersecting pathways will further clarify the complexity of osteogenic regulation and help delineate the precise mechanistic interplay between PACAP and Notch signalling during bone development (Pages 30-31).
Round 3
Reviewer 2 Report
Comments and Suggestions for Authors
Review comments for ijms-3606207.
In this revised manuscript, the authors have demonstrated genuine effort in addressing previous concerns and have provided substantial additional data. However, there are still some tiny concerns, and they are explained below.
- Methodological Clarifications
Several methodological aspects require further clarification to enhance reproducibility. The authors should provide more detailed information about the specific passages used for each cell line, the criteria for cell confluency assessment, and standardized protocols for tissue processing. Additionally, the optimization process for DAPT concentration, while mentioned in the response letter, should be more thoroughly documented in the methods section with appropriate controls and validation data.
- Figure Quality and Presentation
The quality of several figures, particularly the immunocytochemistry images in Figures 5 and 6, could be improved. Higher resolution images with clearer contrast would be better to demonstrate the nuclear translocation of NFATc1. Furthermore, the scale bars and magnifications should be consistent across similar experiments, and quantification methods should be standardized with appropriate controls for antibody specificity.
- Discussion and Mechanistic Models
While the authors have significantly improved the discussion by acknowledging study limitations and proposing mechanistic models, the section would benefit from more nuanced interpretation of the results. The proposed models of compensatory response, cross-modulation, and redundancy should be better integrated with existing literature on PACAP and Notch signaling interactions. Additionally, the authors should more explicitly discuss the translational implications of their findings and potential therapeutic targets.
- Minor Editorial Corrections
Several minor grammatical errors and inconsistencies in terminology remain throughout the manuscript. The authors should carefully proofread the text to ensure clarity and consistency in scientific terminology. Specific attention should be given to the proper use of gene nomenclature and standardization of abbreviations throughout the manuscript.
Author Response
Several methodological aspects require further clarification to enhance reproducibility. The authors should provide more detailed information about the specific passages used for each cell line, the criteria for cell confluency assessment, and standardized protocols for tissue processing. Additionally, the optimization process for DAPT concentration, while mentioned in the response letter, should be more thoroughly documented in the methods section with appropriate controls and validation data.
Thank you for your insightful feedback regarding the methodological aspects of our study. We appreciate your suggestions and are committed to enhancing the clarity and reproducibility of our work. We will expand the documentation of the optimization process for DAPT concentration in the methods section. This will include a comprehensive description of the experimental setup, the range of concentrations tested, the criteria for selecting the optimal concentration, and any controls or validation data that support our findings (pages 33-34).
The quality of several figures, particularly the immunocytochemistry images in Figures 5 and 6, could be improved. Higher resolution images with clearer contrast would be better to demonstrate the nuclear translocation of NFATc1. Furthermore, the scale bars and magnifications should be consistent across similar experiments, and quantification methods should be standardized with appropriate controls for antibody specificity.
Thank you for your valuable feedback regarding the figures in our manuscript, particularly the immunocytochemistry images in Figures 5 and 6. We appreciate your suggestions for improvement and are committed to enhancing the quality of our visual data. We will provide higher resolution images with improved contrast for Figures 5 and 6 to better illustrate the nuclear translocation of NFATc1. This will ensure that the details are more discernible and enhance the overall clarity of the figures. (Figures 5 and 6, Page 41)
While the authors have significantly improved the discussion by acknowledging study limitations and proposing mechanistic models, the section would benefit from more nuanced interpretation of the results. The proposed models of compensatory response, cross-modulation, and redundancy should be better integrated with existing literature on PACAP and Notch signaling interactions. Additionally, the authors should more explicitly discuss the translational implications of their findings and potential therapeutic targets.
Thank you for your insightful feedback. We appreciate your suggestions to enhance the discussion section and will incorporate the following revisions as we will expand our analysis to provide a more detailed interpretation of our findings, highlighting both their strengths and limitations. This will include discussing alternative explanations and the context of our results within the broader field. We will better connect our proposed models of compensatory response, cross-modulation and redundancy with current research on PACAP and Notch signaling interactions. This will involve citing relevant studies that support or contrast with our findings, thereby situating our work within the existing scientific framework. We will explicitly discuss how our findings could inform future therapeutic strategies, including potential targets within the PACAP and Notch pathways. This will help clarify the clinical relevance of our research and suggest avenues for translational research. (Pages 31-32)
Several minor grammatical errors and inconsistencies in terminology remain throughout the manuscript. The authors should carefully proofread the text to ensure clarity and consistency in scientific terminology. Specific attention should be given to the proper use of gene nomenclature and standardization of abbreviations throughout the manuscript.
Thank you for your valuable feedback. We have carefully reviewed the manuscript to address the minor grammatical errors and ensure consistency in scientific terminology as genes are written in italic letters.